# Outsourced Diffusion Sampling:
# Efficient Posterior Inference in Latent Spaces of Generative Models

**Siddarth Venkatraman** [* 1 2]   **Mohsin Hasan** [* 1 2]   **Minsu Kim** [1 2 3]   **Luca Scimeca** [1 2]   **Marcin Sendera** [4]
**Yoshua Bengio** [1 2 5]   **Glen Berseth** [1 2 5]   **Nikolay Malkin** [6]

## Abstract

Any well-behaved generative model over a variable $\mathbf{x}$ can be expressed as a deterministic transformation of an exogenous (*'outsourced'*) Gaussian noise variable $\mathbf{z}$: $\mathbf{x} = f_\theta(\mathbf{z})$. In such a model (*e.g.*, a VAE, GAN, or continuous-time flow-based model), sampling of the target variable $\mathbf{x} \sim p_\theta(\mathbf{x})$ is straightforward, but sampling from a posterior distribution of the form $p(\mathbf{x} \mid \mathbf{y}) \propto p_\theta(\mathbf{x})r(\mathbf{x}, \mathbf{y})$, where $r$ is a constraint function depending on an auxiliary variable $\mathbf{y}$, is generally intractable. We propose to amortize the cost of sampling from such posterior distributions with diffusion models that sample a distribution in the noise space ($\mathbf{z}$). These diffusion samplers are trained by reinforcement learning algorithms to enforce that the transformed samples $f_\theta(\mathbf{z})$ are distributed according to the posterior in the data space ($\mathbf{x}$). For many models and constraints, the posterior in noise space is smoother than in data space, making it more suitable for amortized inference. Our method enables conditional sampling under unconditional GAN, (H)VAE, and flow-based priors, comparing favorably with other inference methods. We demonstrate the proposed *outsourced diffusion sampling* in several experiments with large pretrained prior models: conditional image generation, reinforcement learning with human feedback, and protein structure generation.

## 1. Introduction

Generative models, trained on a dataset to maximize likelihood or related quantities, can become priors for Bayesian

*Table 1.* Common families of generative models can be expressed as deterministic transformations of noise. Here, $d_{\text{latent}}$ refers to the dimension of the latent variable as typically understood in the model; typically, $d_{\text{latent}} \ll d_{\text{data}}$ (§2). Posteriors under these generative model priors are generally intractable, but some models can be fine-tuned by asymptotically unbiased variational objectives to sample intractable posteriors (§3.1). Outsourced diffusion sampling approximates posteriors in the noise space (§3.2 and §4).

| Model | Noise dim. $d_{\text{noise}}$ | Invertible? | Variational tuning? |
|---|---|---|---|
| (H)VAE | $d_{\text{latent}} \cdot N + d_{\text{data}}$ | ✗ | ✓ |
| GAN | $d_{\text{latent}}$ | ✗ | ✗ |
| NF / CNF | $d_{\text{data}}$ | ✓ | ✓/✗ |
| Diffusion | $d_{\text{data}} \cdot (T + 1)$ | ✗ | ✓ |
| Latent diffusion | $d_{\text{latent}} \cdot (T + 1) + d_{\text{data}}$ | ✗ | ✓ |

inference problems. The aim is to approximate or sample from the product of the modeled distribution over a data space with an observation likelihood or other constraint function. While such diverse applications as conditional generation (Dhariwal & Nichol, 2021; Ho & Salimans, 2022), inverse problems (Song et al., 2022; Chung et al., 2023; Venkatraman et al., 2024), and constrained improvement from human feedback (Korbak et al., 2022; Fan et al., 2023) can be cast as posterior inference tasks, sampling from such posterior distributions *when no unbiased target data is available* is generally intractable. For some model families, approximate solutions, such as MCMC, approximate guidance, and variational inference, may be possible. Each of those methods has limitations, such as high cost to reach convergence for multimodal posteriors, intractability of accurate density estimation, and reliance on techniques specialized to the model and constraint.

Fundamentally, generative models are probabilistic programs that produce samples from the distributions they define by a combination of deterministic computation and injection of random noise.[1] This paper argues that the noise space of generative models – the space where the noise injected during generation resides – is an effective target for posterior inference. To be precise, we consider a generative

---

*Equal contribution [1]Mila – Québec AI Institute [2]Université de Montréal [3]KAIST [4]Jagiellonian University [5]CIFAR [6]University of Edinburgh. Correspondence to: <{siddarth.venkatraman, mohsin.hasan}@mila.quebec>.

*Proceedings of the 42$^{nd}$ International Conference on Machine Learning*, Vancouver, Canada. PMLR 267, 2025. Copyright 2025 by the author(s).

---

[1]We refer here to models that produce samples in a bounded number of operations, not to objects such as deep energy-based models, for which sampling is intractable.

*Figure 1.* **Left:** *Top row:* Marginal densities of a CNF that transforms a Gaussian distribution ($t = 0$) to a Swiss roll ($t = 1$). *Middle row:* The constraint function – a mixture of two Gaussians centered an an observation **y** (•) and its reflection through the origin – pulled back to $\mathbf{x}_t$. *Bottom row:* Posterior densities at $\mathbf{x}_t$, proportional to the product of the first two rows. The *rightmost column* shows samples in the data space. **Right:** The same objects shown in noise and data space for a GAN that transforms noise (**z**) to data (**x**). **Outsourced diffusion samplers** approximate $p(\mathbf{x}_0 \mid \mathbf{y})$ or $p(\mathbf{z} \mid \mathbf{y})$, which are smoother than $p(\mathbf{x} \mid \mathbf{y})$ (see Fig. 2).

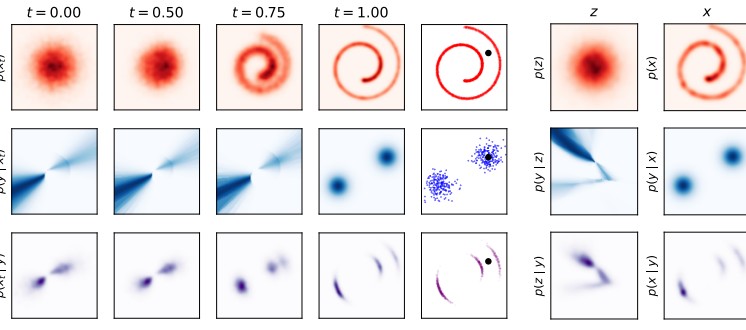

model that expresses data as a deterministic transformation of noise, $\mathbf{x} = f_\theta(\mathbf{z})$, where the noise variable **z** follows a known distribution $\mathbf{z} \sim p_{\mathbf{z}}(\mathbf{z})$ and $\theta$ are the model parameters. This model defines a distribution over **x** – the pushforward $[f_\theta]_* p_{\mathbf{z}}$ of the noise distribution by the deterministic transformation – with density $p(\mathbf{x})$. A constraint function $r(\mathbf{x}, \mathbf{y})$ in the data space, depending on an auxiliary variable **y**, defines a posterior distribution $p(\mathbf{x} \mid \mathbf{y}) \propto p(\mathbf{x}) r(\mathbf{x}, \mathbf{y})$. This posterior can be sampled by inferring a distribution over the noise variable **z** that, when transformed by $f_\theta$, aligns with the posterior in the data space. The posterior in the noise space is often smoother (and lower-dimensional) than the corresponding posterior in the data space (Fig. 1), making it more amenable to efficient sampling (Fig. 2).

While posterior sampling in noise space is still intractable, it can be addressed by methods of black-box variational inference. Recent advances in *diffusion samplers* – diffusion models trained not on a dataset, but to match a given unnormalized density (Zhang & Chen, 2022; Vargas et al., 2023; Richter & Berner, 2024; Sendera et al., 2024) – open an opportunity to model complex posteriors in noise space. We call such amortized posterior inference in the noise space *outsourced diffusion sampling*.

Our exposition and experiments support three claims:

(1) Outsourced diffusion sampling is agnostic to the form of the mapping $f_\theta$ and applicable to a wide range of prior models, including VAEs, GANs, normalizing flows, and continuous-time flow-based models (Table 1).

(2) Outsourced diffusion sampling is an effective posterior inference method under large pretrained generative model priors in a variety of domains: conditional image generation, reinforcement learning with human feedback, discriminator-adjusted GAN sampling, and protein structure generation (Table 2).

(3) Outsourced diffusion sampling is more efficient than amortized inference methods that fit a model to sample the data space posterior directly and than non-amortized methods like MCMC, illustrating the flexibility of diffusion sampling in outsourced noise spaces for sampling

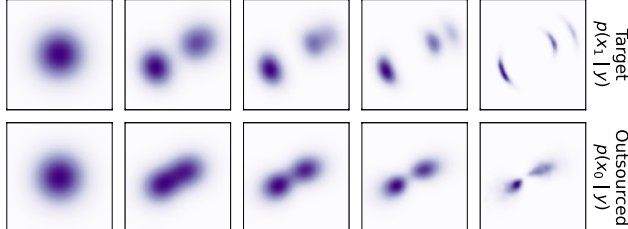

*Figure 2.* Marginal densities of a diffusion sampler of the posteriors from the CNF example in Fig. 1 in data space and noise space. The data space posterior (top row) has well-separated modes and is harder to sample from than the outsourced posterior (bottom row).

complex posteriors (§5).

## 2. Outsourcing Noise in Generative Models

Consider a probabilistic model over a variable **x** taking values in $\mathbb{R}^{d_{\text{data}}}$, with auxiliary latent variables **w** valued in $\mathbb{R}^{d_{\text{latent}}}$. The model is a joint distribution over **x** and **w**, and it induces a distribution over **x**, its marginalization over **w**. In terms of densities (if they exist), if $p(\mathbf{w}, \mathbf{x})$ is the joint density, then the marginal density of **x** is $p(\mathbf{x}) = \int p(\mathbf{w}, \mathbf{x}) \, d\mathbf{w}$.

A form of the *noise outsourcing lemma* (see, *e.g.*, Austin, 2015) states that, under basic assumptions, any such model is equivalent to one augmented with additional latent variables $\mathbf{w}'$, independent of **w** and following a fixed distribution, such that **x** is a (deterministic) function of **w** and $\mathbf{w}'$. In particular, if **w** and $\mathbf{w}'$ are both standard Gaussian, then **x** is a deterministic function of a Gaussian noise variable **z** (the concatentation of **w** and $\mathbf{w}'$), called the *'outsourced'* noise:

**Proposition 2.1** (Noise outsourcing lemma for Gaussians). *Let* **w** *and* **x** *be Borel-measurable random variables valued in* $\mathbb{R}^{d_{\text{latent}}}$ *and* $\mathbb{R}^{d_{\text{data}}}$, *respectively, with* **w** *marginally standard Gaussian, and let* $d_{\text{noise}} > d_{\text{latent}}$. *There exists a random variable* **z** *in* $\mathbb{R}^{d_{\text{noise}}}$ *such that:*

*(1)* **z** *is standard Gaussian;*

*(2)* **w** *is the projection of* **z** *onto its first* $d_{\text{latent}}$ *coordinates;*

*(3) there exists a measurable function* $f : \mathbb{R}^{d_{\text{noise}}} \to \mathbb{R}^{d_{\text{data}}}$ *such that* $(\mathbf{w}, \mathbf{x}) = (\mathbf{w}, f(\mathbf{z}))$ *almost surely.*

Prop. 2.1 ensures that any generative model with marginally Gaussian latent variables can be rewritten as a deterministic function of a higher-dimensional Gaussian noise variable, but does not specify the form of the function $f$ (which is very non-unique). We will be interested in modeling Bayesian posteriors over $\mathbf{x}$ given observations by pulling them back to the noise variable $\mathbf{z}$, using methods agnostic to the form of $f$, which is seen as a black-box transformation.

Common families of generative models have a natural form for $f$, obtainable explicitly from their latent variable structure, that we will exploit. We now explain how several model families, in their basic form, can be expressed as deterministic transformations of noise $\mathbf{z}$ following a Gaussian distribution over $\mathbb{R}^{d_{\text{noise}}}$. See Table 1 for a summary.

**Variational autoencoders (VAEs; Kingma & Welling, 2014).** The generative model in a simple VAE may have the form $\mathbf{x} \sim \mathcal{N}(\boldsymbol{\mu}_\theta(\mathbf{w}), \sigma_\theta^2(\mathbf{w})I_{d_{\text{data}}})$, where $\mathbf{w}$ follows a Gaussian distribution in $\mathbb{R}^{d_{\text{latent}}}$ and $\boldsymbol{\mu}_\theta$ and $\sigma_\theta$ are neural networks outputting a vector and scalar, respectively. This model may be reparametrized as

$$\mathbf{x} = \boldsymbol{\mu}_\theta(\mathbf{w}) + \sigma_\theta(\mathbf{w})\xi, \tag{1}$$

where $\xi \sim \mathcal{N}(0, I_{d_{\text{data}}})$. Thus $\mathbf{x}$ is a deterministic transformation of the concatenation of $\mathbf{w}$ and $\xi$, which follows a Gaussian distribution in $\mathbb{R}^{d_{\text{latent}}+d_{\text{data}}}$. (The encoder, an auxiliary object used in training the VAE, does not form part of the generative model.) **Hierarchical VAEs (HVAEs; Rezende et al., 2014),** generalize VAEs, using a Markovian chain of latent variables in the generative process *i.e.*, a graphical model structure of $\mathbf{w}_N \rightarrow \cdots \rightarrow \mathbf{w}_1 \rightarrow \mathbf{x}$ with each transition a conditional Gaussian distribution. If these variables are all $\mathbb{R}^{d_{\text{latent}}}$-valued, then $\mathbf{x}$ can be similarly reparametrized as a function of the $N$ $d_{\text{latent}}$-dimensional Gaussian noises injected on each transition $\mathbf{w}_{i+1} \rightarrow \mathbf{w}_i$ and the $d_{\text{data}}$-dimensional noise on the last step, as in (1).

**Generative adversarial networks (GANs; Goodfellow et al., 2014).** In a GAN, a generator $G_\theta$ maps Gaussian-distributed noise $\mathbf{z} \sim \mathcal{N}(0, I_{d_{\text{latent}}})$ deterministically to data, $\mathbf{x} = G_\theta(\mathbf{z})$. Thus a GAN is naturally a model with outsourced noise in $\mathbb{R}^{d_{\text{latent}}}$. (The discriminator is an auxiliary object used in training, not a part of the generative model.)

**Normalizing flows (NFs; Rezende & Mohamed, 2015).** In a NF – also naturally a generative model with outsourced noise – the generator $f_\theta$ maps Gaussian noise $\mathbf{z} \sim \mathcal{N}(0, I_{d_{\text{data}}})$ to data $\mathbf{x}$ deterministically, $\mathbf{x} = f_\theta(\mathbf{z})$. Unlike a GAN generator, the function $f_\theta$ is constrained to be invertible and necessarily (in order to model a full-support distribution) must have noise of the same dimension as the data.

**Continuous normalizing flows (CNFs; Chen et al., 2018)** A CNF is an invertible transformation from noise $\mathbf{z} = \mathbf{x}_0$ to data $\mathbf{x} = \mathbf{x}_1$ that is the solution of a neural ordinary

differential equation (ODE)

$$d\mathbf{x}_t = v_\theta(\mathbf{x}_t, t)\, dt.$$

This includes ODEs derived from diffusion models, *e.g.*, DDIMs (Song et al., 2021a), and those trained with flow matching, the family of methods introduced by Lipman et al. (2023); Albergo & Vanden-Eijnden (2023); Liu et al. (2023).

Under regularity conditions on $v_\theta$, a distribution over initial conditions $\mathbf{x}_0$ induces marginal distributions over $\mathbf{x}_t$ for $t > 0$, and in particular over the data variable $\mathbf{x}_1$. The CNF is a generative model with outsourced noise variable $\mathbf{z} = \mathbf{x}_0$.

**Diffusion models.** Diffusion models (Sohl-Dickstein et al., 2015; Ho et al., 2020) and latent diffusion models (Rombach et al., 2021) can also be expressed as deterministic transformations of noise; see Appendix A.1 for discussion and connections.

## 3. Posteriors under Generative Model Priors

For a generative model $p(\mathbf{x})$ of any of the types described in §2, and a positive constraint function $r(\mathbf{x}, \mathbf{y})$ such that $\mathbb{E}_{\mathbf{x} \sim p(\mathbf{x})}[r(\mathbf{x}, \mathbf{y})]$ is finite, we are interested in sampling the posterior distribution $p(\mathbf{x} \mid \mathbf{y}) \propto p(\mathbf{x})r(\mathbf{x}, \mathbf{y})$. Various sources of constraints will be described in §5 (see Table 2).

### 3.1. Posterior Sampling and Approximation

In this section, we describe existing methods for sampling approximately from such intractable posteriors.

**Model-agnostic methods.** The most general methods for sampling from distributions defined by unnormalized densities are Markov chain Monte Carlo (MCMC) methods. These methods may not require fitting parametric models, although hybrid methods – such as adaptive importance sampling (Bugallo et al., 2017), twisted SMC variants (Lawson et al., 2018; 2022), and neural boostrap algorithms (Midgley et al., 2023) – can accelerate their convergence.

MCMC methods are agnostic to the form of the target distribution and are guaranteed to converge to it under mild conditions in the limit of infinite time (or memory in the case of particle filtering methods like SMC (Del Moral et al., 2006; Doucet et al., 2009)), making them anytime algorithms that can trade off computation cost for accuracy. However, MCMCs assume access to the target density and possibly to its gradient, limiting their applicability:

- For **(H)VAEs** and their special case **diffusion models**, the density cannot be computed exactly; only variational bounds are available.
- For **GANs**, the density cannot be computed because the generator is not injective (invertible) and may not even define a full-support distribution over the target space.
- For **CNFs**, the density can be approximated using the

Hutchinson trace estimator (Hutchinson, 1989; Grathwohl et al., 2019), but accurately computing the gradient is expensive, as it requires backpropagating through the computation graph of a neural ODE integrator.

**Monte Carlo methods in latent space.** MCMC sampling can be performed at intermediate time points of continuous-time flow-based models (Cabezas et al., 2024). MCMC techniques are also used for sampling in GAN latent spaces for discriminator-guided sampling (Che et al., 2020; Hou et al., 2025) and for conditional generation by sampling in intermediate activation spaces (Nguyen et al., 2017). Similar approaches are applied to normalizing flows (Coeurdoux et al., 2024). Monte Carlo techniques are also used in diffusion models (Appendix A.1).

**Amortized inference and fine-tuning.** For some families of models, it is possible to train a model that, at convergence to the global optimum, samples from the posterior distribution exactly. This is a problem of variational inference: the model is trained – or perhaps fine-tuned using the prior model as initialization – to be close to the target distribution in some measure of divergence.

For **CNFs**, a method for fine-tuning the drift function $v_\theta$ to yield a CNF that samples from the posterior distribution, known as *adjoint matching*, has recently been proposed by Domingo-Enrich et al. (2024). While this method is asymptotically unbiased, it requires access to the gradient of the likelihood function. Furthermore, it is only applicable to a narrow class of flow-based models, namely, those that are trained from certain marginal couplings and interpolants and closely related to the probability flow ODEs of diffusion models. This restrictiveness is due to adjoint matching converting the neural ODE to an equivalent neural SDE[2], which is not possible in general (*e.g.*, for flow-based models trained using minibatch optimal transport couplings (Tong et al., 2024; Pooladian et al., 2023) or with non-Gaussian source distributions). Naïvely applying adjoint matching to such CNFs gives biased results (Fig. 3).

### 3.2. Bayesian Posterior in Noise Space

We describe how posterior distributions can be pulled back to the noise space of a generative model expressed as a deterministic transformation of an outsourced variable. This relies on a basic measure-theoretic fact regarding the transformation of density functions under pushforward measures:

**Proposition 3.1.** *Suppose that $(Z, \Sigma_Z)$ and $(X, \Sigma_X)$ are measurable spaces and $f : Z \to X$ is measurable. If $\mu$ is a $\sigma$-finite measure on $Z$ and $\nu$ is a $\sigma$-finite measure on $X$*

---

[2]In diffusion models (SDEs), the prior model can also be fine-tuned to sample from the posterior using objectives closely related to those proposed here; see Appendix A.1.

*with $\nu \ll f_*\mu$, then $f_* \left( \left( \frac{d\nu}{df_*\mu} \circ f \right) \cdot \mu \right) = \nu$.*

*In particular, if $\mu$ is a probability measure and $h : X \to \mathbb{R}_{\geq 0}$ is $f_*\mu$-integrable, then $\lambda := \frac{1}{\int h\, d(f_*\mu)} (h \circ f) \cdot \mu$ is a probability measure on $Z$, and $f_*\lambda = \frac{1}{\int h\, d(f_*\mu)} (h \cdot f_*\mu)$ is a probability measure on $X$.*

(See Appendix A.2 for the proof.) In common terms, in terms of densities, the relevance of the proposition of our setting is as follows. Let $f : \mathbb{R}^{d_{\text{noise}}} \to \mathbb{R}^{d_{\text{data}}}$ be a function from the noise space to the data space. A prior density $p(\mathbf{z})$ in the noise space, transformed via $\mathbf{x} = f(\mathbf{z})$, defines a prior $p(\mathbf{x})$ (a density with respect to some reference measure, such as the volume measure on the image of $f$). If $h(\mathbf{x}) = r(\mathbf{x}, \mathbf{y})$ is a constraint function with $\int r(\mathbf{x}, \mathbf{y}) p(\mathbf{x})\, d\mathbf{x} < \infty$, then if $\mathbf{z}'$ is a variable in the noise space distributed with density proportional to $p(\mathbf{z}') r(f(\mathbf{z}'), \mathbf{y})$, and $\mathbf{x}' = f(\mathbf{z}')$, then $\mathbf{x}'$ is distributed with density proportional to $p(\mathbf{x}') r(\mathbf{x}', \mathbf{y})$ in the data space. This means that to sample the posterior in latent space, we can sample from the posterior in noise space (with density $p(\mathbf{z}) r(f(\mathbf{z}), \mathbf{y})$) and transform the sample by $f$. (Note that such sampling does not require computation of the pushforward density. Indeed, $f$ need not be injective or smooth, as would typically be required for such computations.)

**Amortizing outsourced posterior sampling.** Although noise space posteriors might be simpler than the distribution in target space (Figs. 1 and 2), they can still be multimodal and high-dimensional. MCMC methods have been used to sample from noise spaces of NFs and GANs (Che et al., 2020; Cannella et al., 2021), but suffer from long mixing times. In addition, many MCMC methods assume that the target density $p(\mathbf{z}) r(f(\mathbf{z}), \mathbf{y})$ is (efficiently) differentiable, which is not the case when $f$ is a CNF.

Instead, it can be desirable to use amortized variational inference to fit a fast sampler to the latent posterior, that is, to approximate it by a parametric model. We have no samples from this posterior, but have access to its unnormalized density $R(\mathbf{z} \mid \mathbf{y}) := p(\mathbf{z}) r(f(\mathbf{z}), \mathbf{y})$.

We call such a model an *outsourced sampler*, a name motivated by the fact that the factors of variation in the posterior are 'outsourced' to the noise space via the pullback operation. We shall use diffusion models as the variational family, as will be discussed in §4.2.

## 4. Outsourced Diffusion Sampling

### 4.1. Diffusion Samplers for Amortized Inference

*Diffusion sampling* is the variational inference problem of approximating a distribution over $\mathbb{R}^d$, with a given unnormalized density $R : \mathbb{R}^d \to \mathbb{R}_{>0}$, by a diffusion model. Samples from the target distribution, which has density

$p_{\text{target}}(\mathbf{z}) = \frac{1}{Z} R(\mathbf{z})$, are not available, nor do we have access to the normalizing constant $Z$; however, we have the ability to query for the unnormalized density $R(\mathbf{z})$ at any point $\mathbf{z}$. The goal is to train a neural stochastic differential equation

$$\mathbf{z}_0 \sim \mathcal{N}(0, I_d) \quad d\mathbf{z}_t = u_\phi(\mathbf{z}_t, t) \, dt + \sigma_t \, d\mathbf{B}_t, \qquad (2)$$

where $u_\phi$ is a neural network, $\sigma_t$ is a scalar function of time, and $\mathbf{B}_t$ is standard Brownian motion, so that the induced distribution over $\mathbf{z}_1 = \mathbf{z}$ is close to the target distribution $p_{\text{target}}(\mathbf{z})$ in some measure of divergence. (Note that, unlike for diffusion models trained from data, it is standard for generation to proceed in increasing time (from noise at $t = 0$ to the target at $t = 1$.) The model can be sampled by simulating (2) in a time discretization, *e.g.*, using the Euler-Maruyama method.

Training diffusion samplers is more difficult than training typical diffusion models, *i.e.*, maximizing a variational bound on log-likelihood of a dataset. Various objectives have been proposed, including: (1) ones that rely on differentiable simulation of the generative process during training (Li et al., 2020; Kidger et al., 2021; Zhang & Chen, 2022; Vargas et al., 2023) and are linked with optimal control (Berner et al., 2024; Vargas et al., 2024), (2) ones using biased but asymptotically consistent Monte Carlo estimates of the score function (Vargas et al. (2022); Huang et al. (2024); Akhound-Sadegh et al. (2024)), and (3) 'off-policy' divergences that can be optimized on arbitrary generative trajectories not necessarily sampled from the current iteration of the model (Richter et al., 2020; Nüsken & Richter, 2021; Sendera et al., 2024, *inter alia*). A unifying perspective on these methods and analysis in the continuous-time limit were recently given by Berner et al. (2025).

In this work, we adopt training methods of the third kind, using off-policy divergences, as they have have two notable advantages. First, they treat the target $R$ as a black-box reward function and do not require access to the score $\nabla \log R$, as differentiable simulation methods and some Monte Carlo methods do. Second, they can be trained off-policy, on trajectories obtained through exploration, allowing the flexible use of exploration strategies and thus promoting mode discovery, as demonstrated for continuous-space samplers by Malkin et al. (2023); Sendera et al. (2024); Phillips & Cipcigan (2024); Kim et al. (2025). The exact form of the off-policy divergence will be described in §4.2.

### 4.2. Diffusion Sampling in Noise Space

Diffusion models are an attractive choice of variational family due to their ability to sample from complex high-dimensional distributions. We train *outsourced diffusion samplers* by using the methods introduced in §4.1 to approximate posteriors in outsourced noise spaces (§3.2).

The target density we wish to sample takes the form $R(\mathbf{z} \mid$

$\mathbf{y}) \coloneqq p(\mathbf{z}) r(f(\mathbf{z}), \mathbf{y})$. The sampler can be conditioned on the auxiliary variable $\mathbf{y}$ (taking it as an input, resulting in amortization over $\mathbf{y}$ and the possibility of generalizing to new $\mathbf{y}$) or can be trained for a single, fixed value of $\mathbf{y}$ to sample the corresponding posterior.

**Training objective.** To train a diffusion model to sample from the target density $R(\mathbf{z} \mid \mathbf{y})$, we use the trajectory balance objective (TB; Malkin et al., 2022). TB was first introduced in the context of (discrete) generative flow networks (GFlowNets; Bengio et al., 2021; 2023) and generalized to the continuous setting in Lahlou et al. (2023); it is also a close relative of the VarGrad objective (Richter et al., 2020). It was comprehensively evaluated for diffusion samplers in Sendera et al. (2024) and its asymptotic consistency in the continuous-time limit was established in Berner et al. (2025). We briefly review the TB objective and refer to those works for further details.

A diffusion model – a neural network with parameters $\phi$ and possibly conditioned on $\mathbf{y}$ – defines a Markovian distribution over denoising trajectories $\tau = (\mathbf{z}_0 \to \mathbf{z}_{\Delta t} \to \cdots \to \mathbf{z}_1)$, where $\Delta t = \frac{1}{T}$ is the time step of the discretization of the SDE (2), via

$$p_F^\phi(\tau \mid \mathbf{y}) = p(\mathbf{z}_0) \prod_{i=1}^{T} p_F^\phi(\mathbf{z}_{i\Delta t} \mid \mathbf{z}_{(i-1)\Delta t}, \mathbf{y}). \qquad (3)$$

Here $p(\mathbf{z}_0)$ is the density of a fixed distribution over the initial noise (recall that generation goes forward in time) and $p_F^\phi$ is the density of the transition kernel defined by the model, *i.e.*, the probability of transitioning from a sample at a given noise level to a sample at the next-lowest noise level. Similarly, the (fixed) noising process defines a distribution over noising trajectories conditioned on their terminal endpoint: $p_B(\tau \mid \mathbf{z}_1) = \prod_{i=1}^{T} p_B(\mathbf{z}_{(i-1)\Delta t} \mid \mathbf{z}_{i\Delta t})$. The TB objective associated with a trajectory $\tau$ is a squared log-ratio:

$$\mathcal{L}_{\text{TB}}(\tau; \mathbf{y}, \phi) = \left( \log \frac{Z_\phi(\mathbf{y}) p_F^\phi(\tau \mid \mathbf{y})}{R(\mathbf{z}_1 \mid \mathbf{y}) p_B(\tau \mid \mathbf{z}_1)} \right)^2, \qquad (4)$$

where $Z_\phi$ is a learned model that, at optimality, estimates the partition function $\int R(\mathbf{z} \mid \mathbf{y}) \, d\mathbf{z}$. This objective aims to match two distributions over trajectories: the one defined by the denoising model and that defined by the target distribution and the noising kernel. If the two distributions are equal, then their marginal densities at $t = 1$ also coincide.

For training, one draws trajectories $\tau$ from some training distribution (which is not necessarily the current model $p_F^\phi$) and optimizes (4) with respect to the parameters $\phi$. If the TB loss is optimized to 0 for *every* trajectory $\tau$ in the continuous-time limit, the model $p_F^\phi$ asymptotically samples from the target density $R(\mathbf{z} \mid \mathbf{y})$ (Berner et al., 2025). The training loop is described in Algorithm 1.

*Figure 3.* **(a)** Flow paths of a CNF trained with OT-CFM (Tong et al., 2024) from a source distribution (●) to the '2 moons' distribution (●). The source is Gaussian (top row) or a mixture of 8 Gaussians (bottom row). **(b)** The constraint is constructed such that the posterior is the lower moon. CNF flow paths from the lower moon (target posterior) to the source latents (outsourced posterior). **(c)** Flow paths from naive application of Adjoint Matching (Domingo-Enrich et al., 2024), which is is biased for OT flows and non-Gaussian sources. **(d)** Flows starting at samples from an outsourced diffusion model, which samples the latent posterior, give target samples close to the ground truth.

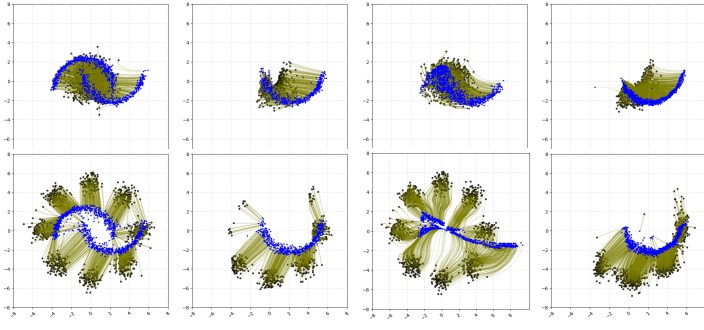

(a) OT-CFM prior   (b) True Posterior   (c) Adj. Matching   (d) Ours

---

**Algorithm 1** Training loop for Outsourced Diffusion Sampling

---

1: **Initialize:** deterministic prior function $f$, randomly initialized noise posterior model $p_F^\phi$, randomly initialized $Z^\phi(\mathbf{y})$, VP-SDE backward policy $p_B$, log reward function $\log r(\mathbf{x}, \mathbf{y})$, on-policy update fraction $p$.
2: **for** each step $n = 1, 2, \ldots, N$ **do**
3:     Sample a batch of trajectories: $\{\tau^{(i)}\}_{i=1}^B \sim p_F^\phi(\tau \mid \mathbf{y})$
4:     **for** $i = 1, \ldots, B$ **do**
5:         Compute log density:
$$\log R^{(i)} \leftarrow \log \mathcal{N}(\mathbf{z}^{(i)}; \mathbf{0}, \mathbf{I}) + r\left(f(\mathbf{z}^{(i)}), \mathbf{y}\right)$$
6:         Store experience $(\tau^{(i)}, \log R^{(i)})$ in buffer $\mathcal{D}$
7:     **end for**
8:     Draw $u \sim \text{Uniform}(0, 1)$
9:     **if** $u \leq p$ **then**
10:         Keep on-policy batch $\{(\tau^{(i)}, \log R^{(i)})\}_{i=1}^B$
11:     **else**
12:         Sample off-policy batch $\{(\tau^{(i)}, \log R^{(i)})\}_{i=1}^B \sim \mathcal{D}$
13:     **end if**
14:     Compute $\mathcal{L}_{\text{TB}}(\tau; \mathbf{y}, \phi)$ for batch using Equation 4.
15:     Update $p_F^\phi$, $Z^\phi(\mathbf{y})$ using $\nabla_\phi \mathcal{L}_{\text{TB}}(\tau; \mathbf{y}, \phi)$.
16: **end for**

---

**Exploration and credit assignment techniques.** We borrow a number of off-policy exploration techniques (such as replay buffers), as well as methods to make training more stable (such as temperature annealing) from the diffusion samplers literature. For details, see Appendix B.

# 5. Experiments

The goal of our experiments is to demonstrate the general applicability of outsourced sampling, highlighting tasks which lack specialized techniques for posterior inference. We list the different tasks, alongside the sources of priors, constraints, and the dimension of noise sampled by the outsourced diffusion model in Table 2.

## 5.1. Class-Conditional Sampling

**Setup.** The prior model $p_\theta(\mathbf{x})$ is an off-the-shelf unconditional image generator trained on the CIFAR-10 dataset (Krizhevsky, 2009). Using a CIFAR-10 classifier $p(\mathbf{y} \mid \mathbf{x})$, we train a posterior class-conditioned generative model $p(\mathbf{x} \mid \mathbf{y}) \propto p_\theta(\mathbf{x})p(\mathbf{y} \mid \mathbf{x})$. For our experiments we work with two priors which achieve high fidelity unconditional generation: a flow matching (CNF) model trained with independent coupling and linear interpolants (I-CFM; Tong et al., 2024), and a spectrally normalized GAN (SN-GAN; Miyato et al., 2018). We use a 13-layer VGG-net model as the classifier (Simonyan & Zisserman, 2015). We compare outsourced diffusion sampling against a powerful MCMC baseline and a recent method for fine-tuning flow-based models:

- **Hamiltonian Monte Carlo (HMC; Brooks et al., 2011)** applied in the noise spaces of both the SN-GAN and CFM priors to sample the outsourced posterior. We highlight that HMC with CNF priors can be quite slow, since it requires differentiating through the ODE integrator.
- **Adjoint Matching (Domingo-Enrich et al., 2024):** see §3.1 and Fig. 3 for discussion. Note that adjoint matching requires access to the gradient of the constraint function.

For the SN-GAN prior, we train an outsourced diffusion model to sample generator noise $\mathbf{z} \in \mathbb{R}^{512}$. For the CFM prior, we sample the initial noise latent at $t = 0$, where $\mathbf{z} \in \mathbb{R}^{3 \times 32 \times 32}$. See details in Appendix B.2.

**Results.** We report average log-reward (classifier log-likelihood) of samples and FID scores (computed with the dataset images of the given class) in Table 3. The CFM posteriors consistently outperform SN-GAN posteriors, reflecting the superior quality of the CFM prior. Among the methods evaluated, adjoint matching with the CFM prior achieves the best performance – as expected, since it is specifically designed for fine-tuning I-CFM models with a Gaussian source (samples in Fig. 8). Outsourced diffusion, a more general approach, also delivers strong conditional generation (visualized in Fig. 4). Unlike adjoint matching and HMC, outsourced diffusion does not rely on gradient information from either the classifier or the prior. Additionally,

*Table 2.* The priors and constraints studied in §5. Outsourced diffusion sampling works in noise spaces of a wide range of generative models and is agnostic to their specific form.

| Task | Constraint | Prior | Prior type | $d_{\text{noise}}$ | $d_{\text{data}}$ |
|------|-----------|-------|-----------|----------|---------|
| CIFAR-10 classifer guidance | CIFAR-10 classifer | SN-GAN | GAN | 128 | $3 \times 32 \times 32$ |
| | | I-CFM | CNF | $3 \times 32 \times 32$ | $3 \times 32 \times 32$ |
| FFHQ text conditioning | ImageReward | StyleGAN3 | GAN | 512 | $3 \times 256 \times 256$ |
| | | NVAE | Hierarchical VAE | $4 \times 20 \times 8 \times 8$ | $3 \times 256 \times 256$ |
| Text-to-Image model RLHF | ImageReward | Stable Diffusion 3 | Latent-CNF | $16 \times 64 \times 64$ | $3 \times 512 \times 512$ |
| Protein structure | Structure Diversity | FoldFlow 2 | Riemannian CNF | $7 \times 64$ | $7 \times 64$ |

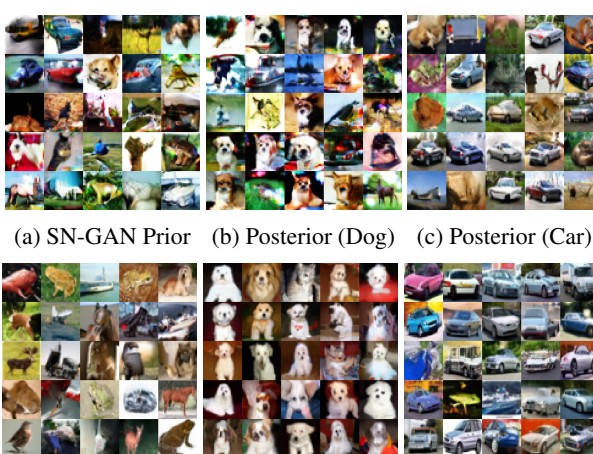

(a) SN-GAN Prior  (b) Posterior (Dog)  (c) Posterior (Car)

(d) I-CFM Prior  (e) Posterior (Dog)  (f) Posterior (Car)

*Figure 4.* CIFAR-10 samples generated using SN-GAN (top row) and CFM (bottom row) priors and posterior samples from trained outsourced diffusion models for the 'Dog' and 'Car' classes.

the outsourced diffusion model offers significant advantages in training efficiency, requiring approximately 5 hours on an A100 GPU compared to 12 hours for adjoint matching (see Appendix C.2). Moreover, we show how we can distill one-step outsourced diffusion samplers in Appendix C.1, with no performance degradation and significant sampling speed advantage.

### 5.2. Conditional High-Resolution Face Generation

**Setup.** Given a generator of high-resolution ($256 \times 256$) human face images trained on the FFHQ dataset (Karras et al., 2021b) as the prior $p_\theta(\mathbf{x})$, we aim to generate faces aligned with a specified text caption $\mathbf{y}$. To achieve this, we use a constraint function given by ImageReward (Xu et al., 2023), a text-image reward model built on the BLIP backbone (Li et al., 2022) that scores images based on their alignment with the provided text prompt and aesthetic quality. The ImageReward score serves as the log-constraint function $\log r(\mathbf{x}, \mathbf{y})$ in our formulation, enabling us to frame the text-conditional face generation problem as posterior inference. For the prior models, we employ NVAE (Vahdat

*Table 3.* CIFAR-10 posterior sampling results for GAN and CNF priors. We report expected classifier log probability and FID scores for the class posteriors, averaged over all 10 classes.

| Prior | Sampler | $\mathbb{E}[\log p(\mathbf{y} \mid \mathbf{x})](\uparrow)$ | FID ($\downarrow$) |
|-------|---------|---------------------|---------|
| SN-GAN | Prior | $-5.37$ | 97.14 |
| | Latent HMC | $-3.26$ | 75.33 |
| | **Outsourced Diff.** | $-3.84$ | 68.12 |
| I-CFM | Prior | $-5.88$ | 84.79 |
| | Latent HMC | $-2.80$ | 46.69 |
| | Adj. Matching | $-3.09$ | 19.45 |
| | **Outsourced Diff.** | $-3.35$ | 34.28 |

& Kautz, 2020) and StyleGAN3 (Karras et al., 2021a), both of which achieve high-fidelity unconditional generation.

NVAE is a deep hierarchical VAE with a large number of latents of different scales. Vahdat & Kautz (2020, Appendix B.6) notes that almost all the feature variance is captured by the first 4 levels of the latent hierarchy. We train the outsourced diffusion model to sample noise $\mathbf{z} \in \mathbb{R}^{4 \times 20 \times 8 \times 8}$ for these levels, which turns out to be sufficient for conditional generation. For StyleGAN3, we sample the generator noise space $\mathbf{z} \in \mathbb{R}^{512}$. Due to the absence of specialized variational techniques for posterior inference with GANs and VAEs, we use HMC sampling of the outsourced noise, targeting the same distribution as the outsourced diffusion sampler, as the baseline. More details in Appendix B.3.

**Results.** We report average ImageReward score and diversity, measured as average cosine distance of CLIP (Radford et al., 2021) embeddings for 100 generated images from the posterior, for 4 different prompts in Table 4. We find HMC for StyleGAN3 can get stuck in bad reward modes, but sometimes obtains high reward. HMC is consistently poor with the NVAE prior, which we attribute to a combination of dimensionality and high energy barriers. Outsourced diffusion samplers consistently generate prompt-accurate posterior samples. Illustrative samples are displayed in Fig. 5 and more uncurated samples in Appendix D. StyleGAN3 posteriors are of higher quality than NVAE posteriors, likely

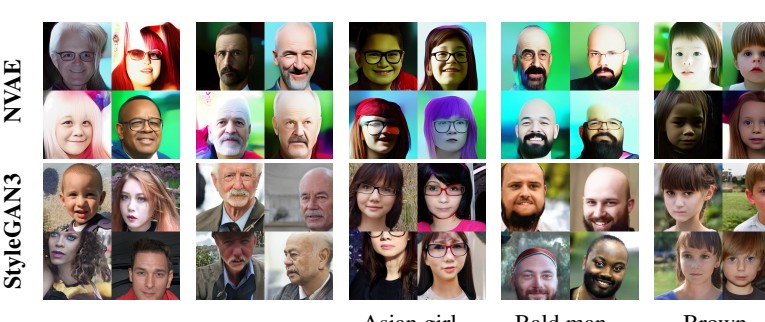

Figure 5. Sampled images from the FFHQ priors and outsourced diffusion posteriors for different prompts. More examples in Appendix D.

Table 4. FFHQ text-conditioning results for NVAE and StyleGAN3 priors. We report average log-reward and CLIP cosine distance (diversity) for posteriors, averaged over the prompts shown in Fig. 5.

| Prior | Sampler | $\mathbb{E}[\log r(\mathbf{x}, \mathbf{y})](\uparrow)$ | CLIP Diversity ($\uparrow$) |
|---|---|---|---|
| NVAE | Prior | −1.94 | 0.30 |
| | Latent HMC | −1.2 | 0.30 |
| | **Outsourced Diff.** | 0.98 | 0.26 |
| StyleGAN3 | Prior | −1.52 | 0.36 |
| | Latent HMC | −0.62 | 0.31 |
| | **Outsourced Diff.** | 1.23 | 0.26 |

because the prior is also stronger.

### 5.3. Text-to-Image RLHF

**Setup.** Diffusion and flow matching models that generate images conditioned on textual prompts often struggle with complex prompts that involve compositional relationships. A promising strategy to address this limitation is to fine-tune such models using reward functions that quantify image-caption alignment. In previous work, Fan et al. (2023); Venkatraman et al. (2024) used Stable-Diffusion-1.5 (Rombach et al., 2021) as a caption-conditioned prior $p(\mathbf{x} \mid \mathbf{y})$ and ImageReward as the unnormalized log-likelihood $\log r(\mathbf{x}, \mathbf{y})$. In both of those works, the prior diffusion model was tuned to sample approximately from the posterior $p^{\text{aligned}}(\mathbf{x} \mid \mathbf{y}) \propto p(\mathbf{x} \mid \mathbf{y}) r(\mathbf{x}, \mathbf{y})$. In our experiments, we instead align Stable Diffusion 3 (SD3; Esser et al., 2024), which is a CNF, not a diffusion model, and is thus unsuitable for fine-tuning using the mentioned techniques.

We train outsourced diffusion samplers of the CNF's $16 \times 64 \times 64$-dimensional noise space. Gradient-based posterior inference techniques, such as adjoint matching and HMC, are prohibitive for a flow model at the scale of SD3, since they involve differentiating through the reward model, high resolution multiscale decoder, and ODE integrator. Instead, as a baseline, we tune the classifier-free-guidance weight (Ho & Salimans, 2022) individually for each prompt and report the best performance. See Appendix B.4 for details.

**Results.** We report the ImageReward score and the average CLIP cosine distance averaged over 4 prompts in Table 5. We present illustrative examples in Fig. 6, with further uncurated samples provided in Appendix E. We find that latent sampling greatly improves reward (and our qualitative assessments) compared to the prior. These results, along with the analysis in §5.2, demonstrate the effectiveness of our proposed method to fine-tune high-dimensional image priors.

Table 5. RLHF finetuning results for SD3 prior. We report expected log reward and CLIP cosine distance (diversity) for posteriors, averaged over the prompts listed in Fig. 6.

| Sampler | $\mathbb{E}[\log r(\mathbf{x}, \mathbf{y})](\uparrow)$ | CLIP diversity ($\uparrow$) |
|---|---|---|
| Prior | 0.791 | 0.19 |
| CFG | 0.84 | 0.17 |
| **Outsourced Diff.** | 1.27 | 0.16 |

A cat and a dog.      A cat riding a llama.

**Prior**   **Posterior**    **Prior**   **Posterior**

Quiet village disrupted by meteor strike.    A human with a horse face and a human with a wolf face.

Figure 6. Sampled images from the SD3 prior and outsourced diffusion posterior for different prompts. More examples in Appendix E.

### 5.4. Protein Secondary Structure Diversity

**Setup.** Many protein generative models have been proposed to tackle the problem of designing novel yet realistic protein structures (Watson et al., 2023; Bose et al., 2024). They learn to produce proteins of $N$ residues by sampling rotations and translations applied to each residue backbone (the space $SE(3)^N$). As our prior $p_\theta(\mathbf{x})$, we use the recent FoldFlow 2 model, which is a Riemannian CNF (on the manifold $SE(3)^N$ embedded in $\mathbb{R}^{7 \times N}$) trained with minibatch OT coupling (Huguet et al., 2024).

Protein residues fold into patterns called secondary struc-

*Table 6.* Results for protein structure experiments. We report the average log-reward, as well as the pairwise TM-Score (as a measure of sample diversity), averaged over 64 samples. Standard deviation over 3 seeds is reported.

| Method | $\mathbb{E}[\log r_{\text{div}}(\mathbf{x})]$ ($\uparrow$) | Pairwise TM-Score ($\downarrow$) |
|---|---|---|
| Prior | $-1.325 \pm 0.014$ | $0.4480 \pm 0.0044$ |
| RW MCMC | $-0.640 \pm 0.073$ | $0.4181 \pm 0.0057$ |
| **Outsourced Diff.** | $0.422 \pm 0.225$ | $0.4407 \pm 0.0706$ |

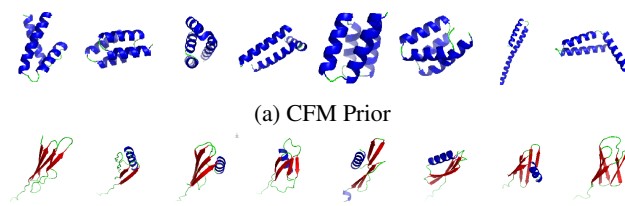

(a) CFM Prior

(b) Outsourced Diffusion Posterior

*Figure 7.* Protein samples with pertinent secondary structures highlighted: $\alpha$-helices (blue), $\beta$-sheets (red), and coils (green).

tures, which include $\alpha$-helices, $\beta$-sheets, and coils. Many protein generative models have issues producing proteins with diverse secondary structures, typically under-sampling proteins with $\beta$-sheets. A natural problem is to produce samples which are both probable under the prior model and exhibit this structural diversity. This can be framed as sampling from $p_\theta(\mathbf{x}) r_{\text{div}}(\mathbf{x})$, where the constraint function $r_{\text{div}}(\mathbf{x})$ assigns high-values to proteins with diverse secondary structures (in-particular, the presence of $\beta$-sheets).

Let $p = [p_\alpha, p_\beta, p_c]$ be a vector representing the proportion of residues that are in $\alpha$-helices, $\beta$-sheets or coils. The particular constraint function that we use, adapted from Huguet et al. (2024), is $r_{\text{div}}(\mathbf{x}) = \frac{e^{-1.2 w^\top p}}{1.2 - \mathcal{H}[p]}$, where $w = [1, 2, 0.5]$ is a weight vector, and $\mathcal{H}[p]$ is the entropy of $p$.

This problem poses two challenges. First, since the model is a Riemannian CNF, the flow ODE cannot be converted into a diffusion SDE, so that adjoint matching and diffusion fine-tuning techniques are not applicable. Second, the constraint function is not differentiable with respect to the generative model's output, ruling out methods such as HMC. We therefore compare our method to a gradient-free MCMC method in the noise space $\mathbf{z}$. We fix the protein length to 64 residues, and evaluate the model achieving the highest diversity score during training. See Appendix B.5 for details.

**Results.** We report the average $\log r_{\text{div}}(\mathbf{x})$, as well as a diversity metric (the pairwise TM-Score) in Table 6. The latter calculates the similarity between pairs of proteins, averaged across a set of generated samples (Zhang & Skolnick, 2004). Uncurated protein samples are shown in Fig. 7. We find that our proposed method samples diverse protein structures rich in $\beta$-sheets more frequently than the baselines, while maintaining a TM-Score comparable to the prior.

## 6. Conclusion

We have proposed outsourced diffusion samplers for efficient posterior inference in the noise spaces of generative models. These samplers take advantage of the expressiveness of diffusion models and the flexibility of off-policy training algorithms for black-box target distributions and can be applied to any model that can be written as a deterministic transformation of noise. While we have demon-

strated the effectiveness of this method in a variety of settings, there are many questions for future work. One natural direction is to extend the method to discrete problems, where the noise space is discrete or the transformation involves discretization. Another is to adapt outsourced diffusion sampling to general probabilistic programs, where the generative model includes both stochasticity and nontrivial control flow, and where current inference methods use MCMC sampling in outsourced noise spaces (Dash et al., 2023).

## Acknowledgments

The authors thank Moksh Jain for comments and suggestions in the course of this project, Kirill Tamogashev for comments on a draft of the manuscript, and Prudencio Tossou and Dominique Beaini for helpful discussions and suggestions about the protein conformation experiments.

L.S. acknowledges support from Recursion. The research of M.S. was funded by National Science Centre, Poland, 2022/45/N/ST6/03374. Y.B. acknowledges the support from CIFAR and the CIFAR AI Chair program as well as NSERC funding for the Herzberg Canada Gold medal. G.B. acknowledges funding support from Samsung AI Lab and CIFAR.

The research was enabled in part by computational resources provided by the Digital Research Alliance of Canada (https://alliancecan.ca), Mila (https://mila.quebec), and NVIDIA.

## Impact Statement

Improving reward finetuning strategies for generative models can enhance their usefulness but also carries risks associated with misuse, reward misspecification and unintended generalization. Careful evaluation and responsible deployment are essential as these methods scale.

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

# A. Theory and Method Details

## A.1. Outsourced Sampling for Diffusion Priors

We describe posterior sampling, outsourced posterior inference and diffusion sampling when the prior generative model is itself a diffusion model.

**Monte Carlo methods.** The 'guidance' term in diffusion posteriors – the difference between the scores of the noised prior and posterior distributions – can be estimated by Monte Carlo integration (Song et al., 2023; Cardoso et al., 2024) or using approximations specialized for constraints arising from linear inverse problems (Kawar et al., 2021; Kadkhodaie & Simoncelli, 2021; Song et al., 2022; Chung et al., 2023). Posterior estimation can also be achieved through stochastic optimization (Graikos et al., 2022; Mardani et al., 2024)

Methods related to sequential Monte Carlo, which treat a modification to the denoising transition kernel as a proposal, have also been proposed (Doucet et al., 2022; Dou & Song, 2024; Chen et al., 2025).

**Amortized methods.** Because generation proceeds in a long sequence of sampling steps, and the modes of the posterior are not known *a priori*, these methods use reinforcement learning techniques to discover regions of high posterior density. Asymptotically unbiased methods include ELEGANT (Uehara et al., 2024) and relative trajectory balance (Venkatraman et al., 2024).

**Outsourcing noise in diffusion models.** Generation of data $\mathbf{x} = \mathbf{x}_0$ is modeled as a Markov process $\mathbf{x}_T \to \cdots \to \mathbf{x}_1 \to \mathbf{x}_0$, where $\mathbf{x}_T \sim \mathcal{N}(0, I_{d_{\text{data}}})$ and the transition from $\mathbf{x}_t$ to $\mathbf{x}_{t-1}$ is conditionally spherical-Gaussian. Via the reparametrization trick, the trajectory of latent variables can be expressed as a function of the initial sample $\mathbf{x}_T$ and the $T$ standard Gaussian noises injected at each step of sampling, just as in a VAE. Thus a diffusion model is a generative model with outsourced noise in $\mathbb{R}^{d_{\text{data}} \cdot (T+1)}$ (see (6) below).

Generalizing this setting, a typical **latent diffusion model** chains a diffusion model in a latent space $\mathbb{R}^{d_{\text{latent}}}$, $\mathbf{w}_T \to \cdots \to \mathbf{w}_1 \to \mathbf{w}_0$, with a Gaussian decoder $\mathbf{w}_0 \to \mathbf{x}$ of the same form as a VAE decoder. Combining the two, the data $\mathbf{x}$ is a deterministic transformation of the concatenation of the initial latent variable $\mathbf{w}_T$, the $T$ standard Gaussian noises injected at each step of sampling, and the noise in the final decoder. Thus a latent diffusion model is a generative model with outsourced noise in $\mathbb{R}^{d_{\text{latent}} \cdot (T+1) + d_{\text{data}}}$.

(Note the similarity to the outsourced interpretation of HVAEs above: a diffusion model indeed be understood as a deep hierarchical VAE. However, a diffusion model is also a neural stochastic differential equation (Tzen & Raginsky, 2019; Song et al., 2021b) integrated in discrete time. In this view, in the continuous-time limit, the outsourced noise is a sample of Brownian motion, and indeed an Itô integral is a deterministic transformation of a Brownian noise random variable.)

**Outsourced autoregressive sampling under diffusion priors recovers relative trajectory balance.** Venkatraman et al. (2024) studied the problem of fine-tuning a diffusion model $p_\theta$ – seen as a transition policy $p_\theta(\mathbf{x}_{i\Delta t} \mid \mathbf{x}_{(i-1)\Delta t})$ – to yield a diffusion model $p_\phi^{\text{post}}$ that samples the product of the distribution $p_\theta(\mathbf{x}_1)$ defined by the prior model with a constraint $r(\mathbf{x}_1)$, where the prior and posterior diffusion model share the noising process and standard Gaussian noise distribution $p(\mathbf{x}_0)$. The *relative trajectory balance* (RTB) objective was proposed; for a trajectory $\tau$,

$$\mathcal{L}_{\text{RTB}}(\tau; \phi) = \left( \log \frac{Z_\phi}{r(\mathbf{x}_1)} + \sum_{i=1}^{T} \log \frac{p_\phi^{\text{post}}(\mathbf{x}_{i\Delta t} \mid \mathbf{x}_{(i-1)\Delta t})}{p_\theta(\mathbf{x}_{i\Delta t} \mid \mathbf{x}_{(i-1)\Delta t})} \right)^2, \tag{5}$$

where $Z_\phi$ is a learned scalar (note the resemblance to (4)).

Let $\xi_i$ be the standard Gaussian noise injected in sampling $x_{i\Delta t}$ conditionally on $x_{(i-1)\Delta t}$, so that the prior model can be rewritten as a deterministic function of the noises $\xi_0, \xi_1, \ldots, \xi_T$ via

$$\mathbf{x}_0 = \xi_0, \quad \mathbf{x}_{i\Delta t} = \mathbf{x}_{(i-1)\Delta t} + v_\theta(\mathbf{x}_{(i-1)\Delta t}, (i-1)\Delta t)\Delta t + \sigma_{i\Delta t}\sqrt{\Delta t}\xi_i, \tag{6}$$

where $\mu_\theta$ outputs the drift of the generative SDE. Similarly, let $\mu_\phi^{\text{post}}$ be the drift of the posterior generative SDE. For a trajectory $\tau$ sampled using a sequence of noises $\xi_0, \xi_1, \ldots, \xi_T$ under the prior model, the RTB loss (5) can then be rewritten

in terms of the noises:

$$
\begin{aligned}
\mathcal{L}_{\text{RTB}}(\tau; \phi) &= \left( \log \frac{Z_\phi}{r(\mathbf{x}_1)} + \sum_{i=1}^{T} \log \frac{\mathcal{N}(\mathbf{x}_{i\Delta t} - \mathbf{x}_{(i-1)\Delta t}; \mu_\phi^{\text{post}}(\mathbf{x}_{(i-1)\Delta t}, (i-1)\Delta t)\Delta t, \sigma_{i\Delta t}^2 \Delta t)}{\mathcal{N}(\mathbf{x}_{i\Delta t} - \mathbf{x}_{(i-1)\Delta t}; \mu_\theta(\mathbf{x}_{(i-1)\Delta t}, (i-1)\Delta t)\Delta t, \sigma_{i\Delta t}^2 \Delta t)} \right)^2 \\
&= \left( \log \frac{Z_\phi}{r(\mathbf{x}_1)} + \sum_{i=1}^{T} \log \frac{\mathcal{N}(\xi_i; \mu_{\text{diff}}(\mathbf{x}_{(i-1)\Delta t}, (i-1)\Delta t)\sqrt{\Delta t}/\sigma_{i\Delta t}, I_{d_{\text{data}}})}{\mathcal{N}(\xi_i; 0, I_{d_{\text{data}}})} \right)^2 \\
&= \left( \log \frac{Z_\phi \mathcal{N}(\xi_0; 0, I_{d_{\text{data}}}) \prod_{i=1}^{T} \mathcal{N}(\xi_i; \mu_{\text{diff}}(\mathbf{x}_{(i-1)\Delta t}, (i-1)\Delta t)\sqrt{\Delta t}/\sigma_{i\Delta t}, I_{d_{\text{data}}})}{r(\mathbf{x}_1) \prod_{i=0}^{T} \mathcal{N}(\xi_i; 0, I_{d_{\text{data}}})} \right)^2
\end{aligned}
\tag{7}
$$

where $\mu_{\text{diff}}(\mathbf{x}, t) := \mu_\phi^{\text{post}}(\mathbf{x}, t) - \mu_\theta(\mathbf{x}, t)$.

Consider now an amortized sampler of the outsourced noises $\xi_0, \xi_1, \ldots, \xi_T$ that generates the variables autoregressively ($\xi_0$ from a standard Gaussian, then subsequently each $\xi_i$ from a Gaussian with unit variance and mean conditioned on the previously sampled noises). The transition policy density of this sampler can be written

$$
p_F(\xi_i \mid \xi_0, \ldots, \xi_{i-1}) = \mathcal{N}(\xi_i; \mu_{\text{outsourced}}(\xi_0, \ldots, \xi_{i-1}), I_{d_{\text{data}}}).
$$

Under a model parametrization in which the policy takes as input the intermediate state $\mathbf{x}_{(i-1)\Delta t}$, computed as a function of the noises $\xi_0, \ldots, \xi_{i-1}$ using the prior model,

$$
\mu_{\text{outsourced}}(\xi_0, \ldots, \xi_{i-1}) = \mu_{\text{diff}}(\mathbf{x}_{(i-1)\Delta t}, (i-1)\Delta t)\sqrt{\Delta t}/\sigma_{i\Delta t},
$$

the numerator in (7) is *precisely* $Z_\phi p_F(\tau)$, where $\tau$ is the sampling trajectory of the autoregressive sampler generating $\xi_0, \xi_1, \ldots, \xi_T$. The denominator is $r(\mathbf{x}_1)$ multiplied with the prior density of the noise, which is the target density for the sampler (the unnormalized density of the outsourced posterior).

Thus we see that (7) exactly recovers the trajectory balance objective (Malkin et al., 2022) for an autoregressive sampler of outsourced noise.

### A.2. Proof of Prop. 3.1

**Proposition A.1.** *Suppose that $(Z, \Sigma_Z)$ and $(X, \Sigma_X)$ are measurable spaces and $f : Z \to X$ is measurable. If $\mu$ is a $\sigma$-finite measure on $Z$ and $\nu$ is a $\sigma$-finite measure on $X$ with $\nu \ll f_*\mu$, then $f_*\left(\left(\frac{d\nu}{d f_*\mu} \circ f\right) \cdot \mu\right) = \nu$.*

*In particular, if $\mu$ is a probability measure and $h : X \to \mathbb{R}_{\geq 0}$ is $f_*\mu$-integrable, then $\lambda := \frac{1}{\int h\, d(f_*\mu)}(h \circ f) \cdot \mu$ is a probability measure on $Z$, and $f_*\lambda = \frac{1}{\int h\, d(f_*\mu)}(h \cdot f_*\mu)$ is a probability measure on $X$.*

*Proof of Prop. 3.1.* Let $h = \frac{d\nu}{d f_*\mu}$; we must show that $f_*((h \circ f) \cdot \mu) = h \cdot f_*\mu$. Let $E \in \Sigma_X$ and $D = f^{-1}(E) \in \Sigma_Z$. By definitions and properties of pushforward measures,

$$
\begin{aligned}
f_*((h \circ f) \cdot \mu)(E) &= ((h \circ f) \cdot \mu)(D) \\
&= \int_Z (h \circ f) \mathbf{1}_D \, d\mu \\
&= \int_Z (h \mathbf{1}_E) \circ f \, d\mu \\
&= \int_X h \mathbf{1}_E \, d f_*\mu \\
&= (h \cdot f_*\mu)(E),
\end{aligned}
$$

as required.

For the second part of the proposition, if $\nu = h \cdot f_*\mu$, then $\frac{d\nu}{d f_*\mu} = h$ $f_*\mu$-almost everywhere, and the result follows easily from the first part by linearity of the pushforward. $\square$

# B. Experiment Details

Code for our experiments is available at `https://github.com/HyperPotatoNeo/Outsourced_Diffusion_Sampling`.

## B.1. Diffusion model

For all experiments, we use a convolutional UNet architecture (Ronneberger et al., 2015) for the diffusion model. This architecture is naturally well-suited for latent spaces structured as image feature maps. However, for other types of latent representations, we found that simply reshaping them into $H \times W \times C$ feature maps and treating them as images yielded surprisingly effective results.

We use the variance preserving SDE (Song et al., 2021b) as backward policy $P_B$ for the trajectory balance loss in (4), with a discretisation of 25 steps. For all of the experiments, we use off-policy training with a replay buffer similar to Sendera et al. (2024). Specifically, for every training update we randomly sample either from the replay buffer, or generate new on-policy samples which are added to the replay buffer according to the buffer probability $\alpha$ – whose value specific to experiments is provided in the following sections.

## B.2. Class-Conditional Sampling

We set the classifier inverse temperature $\beta = 4$ for all our baselines, otherwise the soft logits of the classifier results in a fuzzy posterior. We provide some additional information regarding implementation of the baselines below:

**Outsourced Diffusion.** We learn to sample the 128 dimensional generator noise with the outsourced sampler. To keep architectures consistent across experiments, we used the convolutional UNet to sample this noise by reshaping them to $2 \times 8 \times 8$ feature maps. For sampling from the CNF prior, we use 45 Euler steps to discretise the ODE. The outsourced sampler operates on the noise space which the same dimensionality as the output image – $3 \times 32 \times 32$.

**Adjoint Matching.** To fine-tune flow matching models with a classifier, we use adjoint matching following the approach described in Domingo-Enrich et al. (2024). Due to the unavailability of the open source code at this time, we made our own implementation which we found works extremely well. First, we convert the ODE inference of conditional flow matching into a memoryless SDE, ensuring that both have the same marginal distribution as outlined in the referenced paper. Next, we apply the adjoint matching method for the stochastic optimal controls of the memoryless SDE. This process requires gradient information of the log-reward (*i.e.*, the classifier's log-likelihood), which we compute using PyTorch's autograd functionality.

All hyperparameters remain identical to those of the outsourced diffusion sampler, including the learning rate, except for the number of training iterations and the temperature annealing schedule. Training is significantly slower because it requires computing the log-reward gradients and simulating the full trajectory for adjoint matching. Therefore, we set a total of 2,500 iterations with 1,000 linear temperature annealing steps. This configuration results in similar or reduced wall time compared to the outsourced diffusion sampler while achieving stable convergence and high performance on CIFAR-10. Figure 8 shows the qualitative results of applying adjoint matching to the CIFAR-10 task.

Note that for Stable Diffusion 3, the actual implementation of the inference pipeline—which involves a diffusion transformer with various combinations of language embedding fusion for multimodality—is not fully open-sourced. Implementing a memoryless SDE on top of this pipeline is non-trivial and requires careful tuning and integration with the internal processes of Stable Diffusion's inference mechanisms.

**Temperature Annealing.** We anneal the inverse temperature from $\beta = 2$ to the final $\beta = 4$. We tune the schedule linearly over the first $2,000$ steps of training for outsourced diffusion, and $1,000$ steps for adjoint matching.

**HMC baselines.** We use hamiltorch (Cobb & Jalaian, 2021) to implement HMC. We use step size of $10^{-2}$, with 5 leapfrog integration steps. We use a burn-in chain of length 100 before starting to collecting samples, spaced out by 10 samples. Chains are run for $1,000$ samples, after which we reset the seed to help diversity. The runtime of HMC is quite slow with the CNF prior due to gradient computation. A chain of $1,000$ samples takes close to an hour on $A100$ GPUs. We only keep 90 samples from this chain to preserve diversity, but these are still correlated. It would take close to 10 hours to generate $1,000$ samples for FID computation, but using extra resources we can run parallel chains. For comparison, outsourced diffusion takes close to 5 hours for full training, after which sample generation is extremely cheap due to amortization. Despite the

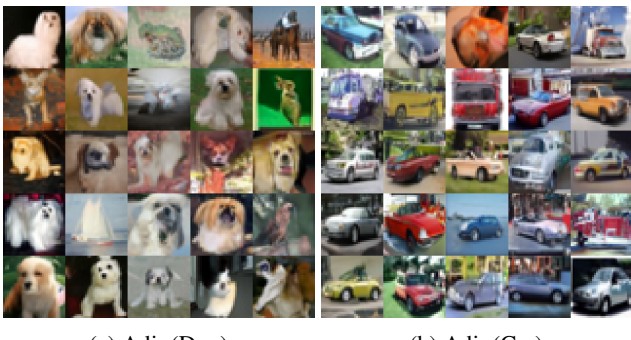

(a) Adj. (Dog)          (b) Adj. (Car)

*Figure 8.* CIFAR-10 samples generated Adjoint Matching (Adj.) for the *Dog* and *Car* classes.

need to train a model, outsourced diffusion is actually more memory-efficient than even a single HMC chain, as it eliminates the need to compute gradients through the ODE integrator. We visualize samples from a latent HMC chain in Fig. 40.

Our results for each class are listed in Table 7 and Table 8. We additionally evaluate the ELBO for I-CFM posteriors in Table 9.

*Table 7.* SN-GAN results on CIFAR-10

| Class | Prior | | HMC | | Outsourced Diff. | |
|---|---|---|---|---|---|---|
| | **Reward** | **FID** | **Reward** | **FID** | **Reward** | **FID** |
| Airplane | -5.82 | 100.28 | -2.53 | 82.56 | -3.89 | 79.55 |
| Car | -5.36 | 125.16 | -3.91 | 94.81 | -4.22 | 85.88 |
| Bird | -5.40 | 71.30 | -4.86 | 52.14 | -3.91 | 59.76 |
| Cat | -5.40 | 68.30 | -2.21 | 67.55 | -3.97 | 55.05 |
| Deer | -5.40 | 70.10 | -3.67 | 51.10 | -3.38 | 43.04 |
| Dog | -5.32 | 95.62 | -4.00 | 82.32 | -4.23 | 68.52 |
| Frog | -4.98 | 93.00 | -4.20 | 68.12 | -3.91 | 74.70 |
| Horse | -5.23 | 97.53 | -3.21 | 64.60 | -3.70 | 52.87 |
| Ship | -5.19 | 116.89 | -2.11 | 94.45 | -3.76 | 79.73 |
| Truck | -5.59 | 133.23 | -2.25 | 97.88 | -3.50 | 82.07 |
| **AVG** | **-5.37** | **97.14** | **-3.26** | **75.33** | **-3.847** | **68.117** |

### B.3. Conditional High-Resolution Face Generation

For all prompts we use fixed inverse temperature $\beta = 100$, which we found to be a suitable reward scale.

**Outsourced Diffusion.** StyleGAN3 uses 512 dimensional generator noise. Similar to our SN-GAN experiment for CIFAR, we reshape this into a $2 \times 16 \times 16$ feature map to be passed to the UNet model. The NVAE prior for FFHQ is a very deep latent variable model, having 36 total latent groups starting from $8 \times 8$ scale all the way up to $128 \times 128$. The joint dimensionality of this latent space is extremely large, making joint posterior inference very challenging. Luckily most variable features of interest are captured in the first 4 latent groups of size $20 \times 8 \times 8$ each. We stack the noise groups to create $80 \times 8 \times 8$ feature maps jointly diffused by the UNet. We use 25 steps for diffusion sampling.

**HMC baseline.** We use $5 \cdot 10^{-3}$ as the step size for StyleGAN3, and $10^{-2}$ as the step size for NVAE. We use 5 leapfrog integration steps. We only require 100 samples for evaluation in this task (unlike 1000 needed for FID in CIFAR), and so we can afford to only collect 2 samples for every 1000 length chain. We collect the samples at $t = 500$ and $t = 1000$. This takes around 3 hours for StyleGAN3 and 5 hours for NVAE without parallelism. We only sample the first 4 latent groups with NVAE prior.

We find that unlike the CIFAR-10 priors, HMC struggles to obtain high reward with these priors, but performs better with StyleGAN3 than NVAE. We attribute the particularly poor performance of NVAE to the high dimensionality and high energy barriers. Interestingly, outsourced diffusion performs significantly better for sampling these posteriors. We suspect that it is

*Table 8.* I-CFM results on CIFAR-10

| Class | Prior | | HMC | | Adj. Matching | | Outsourced Diff. | |
|---|---|---|---|---|---|---|---|---|
| | Reward | FID | Reward | FID | Reward | FID | Reward | FID |
| Airplane | -5.81 | 73.09 | -2.45 | 62.24 | -3.30 | 27.64 | -3.36 | 47.61 |
| Car | -6.22 | 92.07 | -2.12 | 24.85 | -2.93 | 17.51 | -3.02 | 19.12 |
| Bird | -5.94 | 73.48 | -2.93 | 60.75 | -3.16 | 23.09 | -3.11 | 41.04 |
| Cat | -5.53 | 70.33 | -3.60 | 54.32 | -3.33 | 20.11 | -3.57 | 43.66 |
| Deer | -5.59 | 72.19 | -2.72 | 49.46 | -3.08 | 15.52 | -3.78 | 31.87 |
| Dog | -5.83 | 89.38 | -3.89 | 41.13 | -3.21 | 23.11 | -3.57 | 34.33 |
| Frog | -6.06 | 93.08 | -3.36 | 55.64 | -3.27 | 20.86 | -3.98 | 34.74 |
| Horse | -6.13 | 82.88 | -2.10 | 46.22 | -2.79 | 16.59 | -2.95 | 31.85 |
| Ship | -5.85 | 102.33 | -2.28 | 37.80 | -3.09 | 18.00 | -2.92 | 32.13 |
| Truck | -5.82 | 99.10 | -2.55 | 34.48 | -2.55 | 12.34 | -3.20 | 26.46 |
| **AVG** | **-5.88** | **84.79** | **-2.80** | **46.69** | **-3.09** | **19.45** | **-3.35** | **34.28** |

*Table 9.* Additional comparisons against the training-free DPS (Chung et al., 2023) and diffusion fine-tuning with RTB (Venkatraman et al., 2024). For methods where it is applicable, we also report the estimated ELBO.

| Model | Sampler | $\mathbb{E}[\log p(\mathbf{y} \mid \mathbf{x})]$ ($\uparrow$) | FID ($\downarrow$) | ELBO ($\uparrow$) |
|---|---|---|---|---|
| I-CFM | Prior | $-5.88$ | 84.79 | $-24.04$ |
| | DPS | $-2.22$ | 84.96 | - |
| | RTB | $-4.20$ | 90.77 | $-147.69$ |
| | Latent HMC | $-2.80$ | 46.69 | - |
| | Adj. Matching | $-3.09$ | 19.45 | $-17.23$ |
| | **Outsourced Diff.** | $-3.35$ | 34.28 | $-20.36$ |

primarily the nice mode mixing properties of the diffusion annealing path that facilitate this. However, an additional factor might be the benefits of amortization, which is an interesting direction for future work to investigate. We present the results for different prompts in Table 10, and showcase the first 10 samples for each prompt, generated from a fixed random seed, in Appendix D.

*Table 10.* StyleGAN3 results on FFHQ

| Prompt | Prior | | HMC | | Outsourced Diff. | |
|---|---|---|---|---|---|---|
| | Reward | Diversity | Reward | Diversity | Reward | Diversity |
| An old man | -1.02 | 0.35 | -0.60 | 0.3 | 1.62 | 0.32 |
| A young Asian girl with glasses | -1.97 | 0.36 | -0.81 | 0.32 | 1.13 | 0.21 |
| A bald man with a black beard | -1.89 | 0.38 | -0.71 | 0.32 | 1.02 | 0.27 |
| A brown-haired child | -1.2 | 0.36 | -0.35 | 0.28 | 1.14 | 0.23 |
| **AVG** | **-1.52** | **0.36** | **-0.62** | **0.31** | **1.23** | **0.26** |

### B.4. Text-To-Image RLHF

We intentionally choose prompts that pose a challenge for Stable Diffusion 3 while still receiving reliable feedback from ImageReward. Since SD3 is generally a more powerful model than ImageReward, this approach is not applicable to most prompts. However, in a real-world scenario, we anticipate the use of a better preference model trained with human feedback, which would offer more reliable guidance for improving the generative model. For training, we use a fixed inverse temperature $\beta = 30$.

**Outsourced Diffusion.** We learn to sample from the noise space of SD3, which is a latent CNF. This means we can

*Table 11.* StyleGAN3 results on FFHQ

| Prompt | Prior | | HMC | | Outsourced Diff. | |
|---|---|---|---|---|---|---|
| | **Reward** | **Diversity** | **Reward** | **Diversity** | **Reward** | **Diversity** |
| An old man. | -1.74 | 0.30 | -1.24 | 0.32 | 1.38 | 0.31 |
| A young Asian girl with glasses. | -2.12 | 0.30 | -0.95 | 0.31 | 0.70 | 0.25 |
| A bald man with a black beard. | -2.16 | 0.29 | -1.40 | 0.29 | 1.32 | 0.26 |
| A brown-haired child. | -1.76 | 0.30 | -1.21 | 0.31 | 0.53 | 0.23 |
| **AVG** | **-1.94** | **0.30** | **-1.20** | **0.30** | **0.98** | **0.26** |

sample at a reduced dimensionality from the full image (of size $3 \times 512 \times 512$), however the latent space is still fairly high dimensional with shape $16 \times 64 \times 64$.

**Classifier-Free Guidance.** Since SD3 is trained as both an unconditional and conditional model, we can use CFG to approximately sample from lowered temperature conditional distribution:

$$\hat{\mathbf{v}}_\theta(\mathbf{x}_t, \mathbf{y}) = (1 + w)\mathbf{v}_\theta(\mathbf{x}_t, \mathbf{y}) - w\mathbf{v}_\theta(\mathbf{x}_t) \tag{8}$$

Increasing the guidance scale $w$ generally guides the model to be more prompt accurate at the cost of diversity and if increased too much, image fidelity. We find tuning the CFG weight slightly improves score with ImageReward. Increasing $w$ resulted in degraded performance ("A cat riding a llama."), so we report score with the default guidance scale $w = 5.0$ (same as prior).

We present the results for different prompts in Table 12, and showcase the first 10 samples for each prompt, generated from a fixed random seed, in Appendix E. We set $w = (6.0, 5.0, 5.5, 5.5)$ for the prompts ordered as in the table.

Additionally, we train Outsourced Diffusion posteriors using Stable Diffusion 1.5 with the same prompts as in (Venkatraman et al., 2024). We compare our results against those reported in their paper in Table 13.

*Table 12.* SD3 RLHF results for each prompt

| Prompt | Prior | | | CFG | | | Outsourced Diff. | | |
|---|---|---|---|---|---|---|---|---|---|
| | **Reward** | **Diversity** | **ELBO** | **Reward** | **Diversity** | **ELBO** | **Reward** | **Diversity** | **ELBO** |
| A cat and a dog. | 0.50 | 0.14 | 3.55 | 0.61 | 0.10 | - | 1.24 | 0.09 | 27.25 |
| A cat riding a llama. | 0.79 | 0.18 | 1.01 | 0.79 | 0.18 | - | 1.53 | 0.14 | 10.83 |
| A quiet village is disrupted by a meteor strike. | 0.65 | 0.24 | 1.30 | 0.71 | 0.20 | - | 0.94 | 0.21 | 23.20 |
| A human with a horse face and a human with a wolf face. | 1.22 | 0.20 | 19.32 | 1.26 | 0.22 | - | 1.36 | 0.18 | 26.10 |
| **AVG** | **0.79** | **0.19** | **6.29** | **0.84** | **0.17** | **-** | **1.27** | **0.16** | **21.85** |

*Table 13.* SD 1.5 RLHF results, averaged across prompts

| Sampler | $\mathbb{E}[\log r(\mathbf{x}, \mathbf{y})](\uparrow)$ | CLIP diversity ($\uparrow$) |
|---|---|---|
| Prior | -0.17 | 0.18 |
| DDPO | 1.37 | 0.09 |
| DPOK | 1.23 | 0.13 |
| RTB | 1.4 | 0.11 |
| **Outsourced Diff.** | 1.26 | 0.14 |

## B.5. Protein Structure Prediction

The specific form of the constraint function $r_{\text{div}}(\mathbf{x})$ (which can also be thought of as a reward) is adapted from (Huguet et al., 2024). It is a monotonic function of the entropy over secondary structure proportions (restricted to $\alpha$-helices, $\beta$-sheets and coils). The non-linear transformation increases the discrepancy between the rewards of samples with and without $\beta$-sheets. The expression is adjusted so that it is positive, to allow us to take the logarithm.

The $e^{-1.2}$ multiplier ensures the log-constraint function is roughly centered around 0. We found this to improve numerical stability in the diffusion training.

**Outsourced Diffusion.** In the outsourced diffusion experiments, an inverse temperature of $\beta = 400.0$ was used to allow for improvement in this sparse reward setting. The diffusion sampler used 20 sampling steps.

The $SE(3)^N$ elements representing protein structure were parameterized as $\mathbb{R}^{7 \times N}$ vectors, with the first 4 coordinates being a quaternion representation for a rotation matrix and the 3 other numbers representing the translation vector. For the UNet model, the protein coordinates were shaped into $7 \times 8 \times 8$ vectors. Additionally, the diffusion sampler was first pre-trained (for 200 epochs) using the denoising score matching objective (Ho et al., 2020), with samples from $\mathcal{U}(SO(3)^N) \times \mathcal{N}(0, 10^2 I_3)$ (where $\mathcal{U}$ is the Haar measure, *i.e.*, the unique invariant probability measure on $SO(3)^N$), the initial $\mathbf{x}_0$ distribution used for training the CNF prior model. All inference parameters for the flow model were based on the default configuration from the FoldFlow 2 paper, including 50 steps for integrating the flow ODE (Huguet et al., 2024).

For TB training, we used a replay buffer, where 1/4 of the samples are drawn proportional to their reward, and the rest are sampled uniformly. The buffer is used at each iteration with a probability of 0.2 ($\alpha = 0.2$). The reason for the modification from the standard uniform buffer used in other experiments is to make the best use of high reward samples, which are rare especially early on in training. A learning rate of $10^{-5}$, and a batch size of 16 was used. Gradient $l_2$ norms were clipped to 0.05.

Due to policy collapse and training instability, models were saved every 100 training iterations, and the model with highest reward was selected for evaluation. The diffusion model was trained for 4 A100 GPU hours.

**Random Walk MCMC baseline.** A Gaussian proposal, $p(\mathbf{z}' \mid \mathbf{z}) = \mathcal{N}(\mathbf{z}'; \mathbf{z}, 0.01^2 I)$ (*i.e.*, a step-size of 0.01). The quaternion dimensions are normalized to have unit norm (projected back to $SO(3)^N$).

Note this proposal is symmetric. The MCMC chain was run for 1000 iterations, with 32 chains in parallel. Metrics were evaluated on samples from the iterations 900 and 1000 (to reduce sample correlation). This was run for 8 A100 GPU hours.

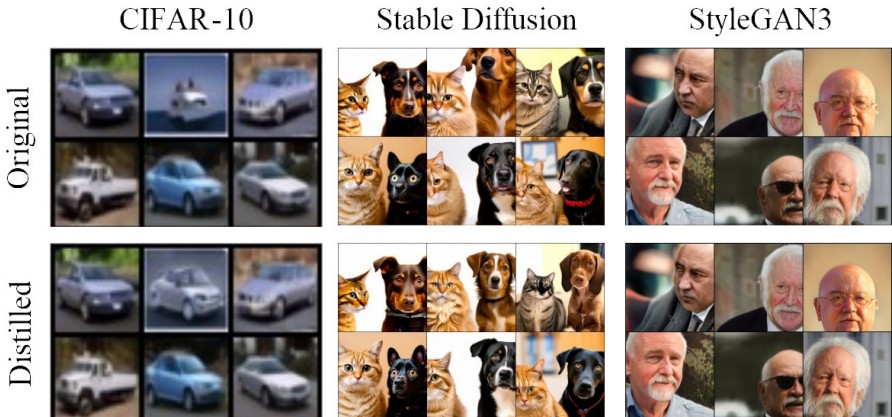

*Figure 9.* Posterior samples via the original outsourced diffusion sampler and the distilled one-step sampler in three experiments. We use the prompt "A cat and a dog." for the stable diffusion experiments, and "An old man" for the SN-GAN experiments.

## C. Additional Experiments

### C.1. Distillation and One-Step Outsourced Samplers

Training outsourced samplers of the kind described in this paper may have the downside of an increase in the number of sampling steps necessary to produce posterior samples of interest. We perform further experiments to show how we can mitigate these side effects via distillation. We distill outsourced models for the CMF posterior on the 'car' class in CIFAR-10, the SN-GAN posterior for "A dog and a cat", and a stable diffusion posterior for a "A green car". We train distilled one-step samplers equivalent in architecture and size to our original diffusion samplers. We use the trained outsourced samplers as *teacher* models and employ a simple training regime whereby we sample $\mathbf{z}_0 \sim \mathcal{N}(0, I_d)$, use an ODE sampling scheme to sample $\mathbf{z}_1$, and then learn a one-step mapping from $\mathbf{z}_0$ to $\mathbf{z}_1$ with the student model. We use Mean Square Error (MSE) loss and a simple variance agnostic regularizer to train the student model and encourage diversity.

*Table 14.* FID scores of standard and distilled outsourced diffusion samplers for a CFM prior on CIFAR-10.

| Sampler | FID |
| --- | --- |
| SDE | 38.65 |
| ODE | 36.71 |
| Distilled | 33.85 |

We report in Fig. 9 original and posterior samples for all experiments. We observe high-quality distilled samples, undistinguishable from the original outsourced diffusion sampler. Furthemore, we report in Table 14 the FID score computed from samples from the original and distilled sampler, with respect to the original CIFAR-10 samples of the class of interest. We observe comparable, if not improved, FID for our distilled model. In line with the results in the main paper, we posit that the ability to fit high fidelity one-step samplers is due to the simplicity of the properties of the target distribution in latent space, often smoother and lower dimensional than the distribution in data space, leading to easy-to-learn transforms from $\mathbf{z}_0$ to $\mathbf{z}_1$.

### C.2. Efficiency Analysis

The results in Appendix C.1 show that the diffusion sampler can be distilled into a single-step generator, enabling inference of the fine-tuned model without incurring additional sampling time. As a result, the inference efficiency remains comparable to that of direct finetuning methods.

On the CIFAR-10 dataset, our method achieves approximately twice the training speed of adjoint matching when using an NVIDIA A100 GPU. This performance advantage is expected to grow for higher-dimensional outputs. It is important to note that the requirement for gradient computation not only raises training memory costs but also limits the flexibility of the method. For tasks where the reward gradient is unavailable, such as many protein or molecule tasks, Adjoint Matching is not applicable. In contrast, the outsourced diffusion sampler can be efficiently applied to arbitrary black-box tasks where the reward (or prior) is not differentiable.

# D. FFHQ Samples

## D.1. Prior.

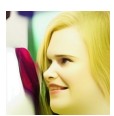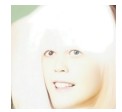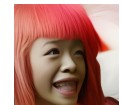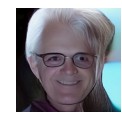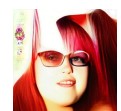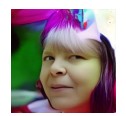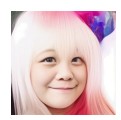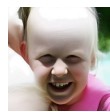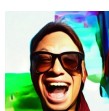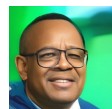

*Figure 10.* NVAE prior

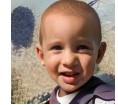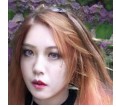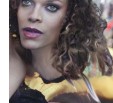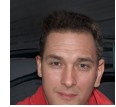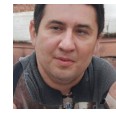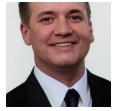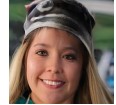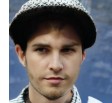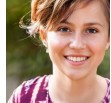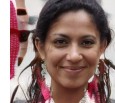

*Figure 11.* StyleGAN3 prior

## D.2. An old man.

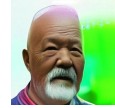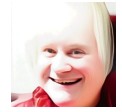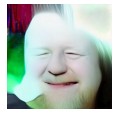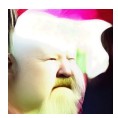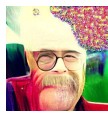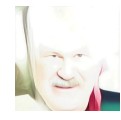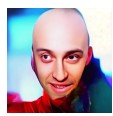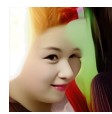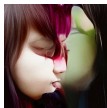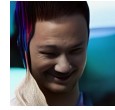

*Figure 12.* NVAE HMC

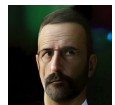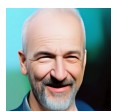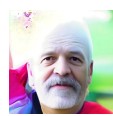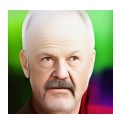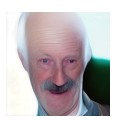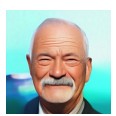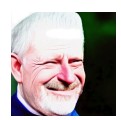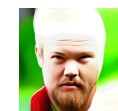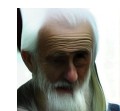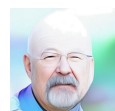

*Figure 13.* NVAE Outsourced Diffusion

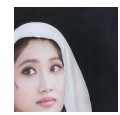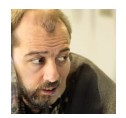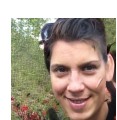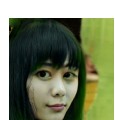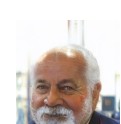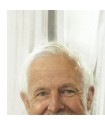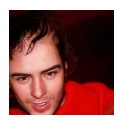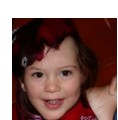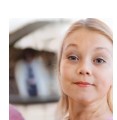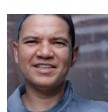

*Figure 14.* StyleGAN3 HMC

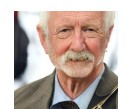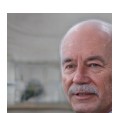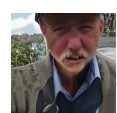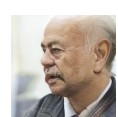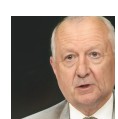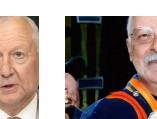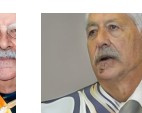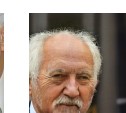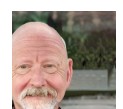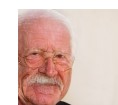

*Figure 15.* StyleGAN3 Outsourced Diffusion

### D.3. An asian girl with glasses.

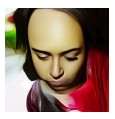 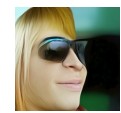 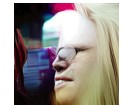 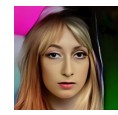 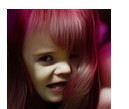 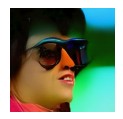 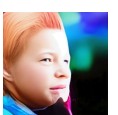 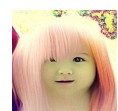 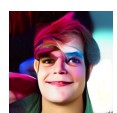 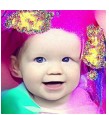

*Figure 16.* NVAE HMC

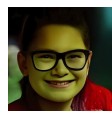 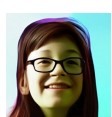 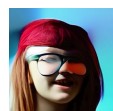 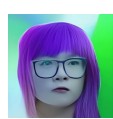 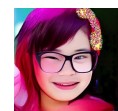 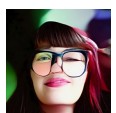 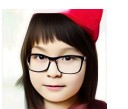 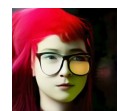 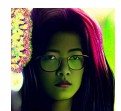 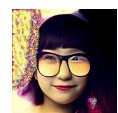

*Figure 17.* NVAE Outsourced Diffusion

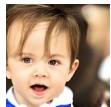 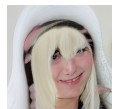 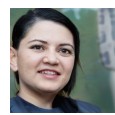 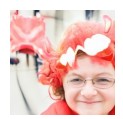 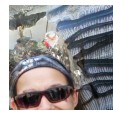 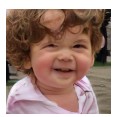 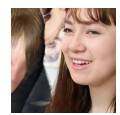 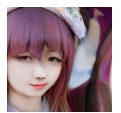 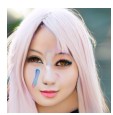 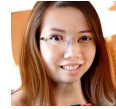

*Figure 18.* StyleGAN3 HMC

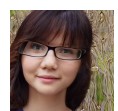 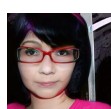 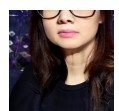 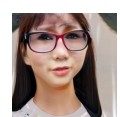 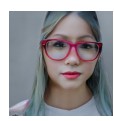 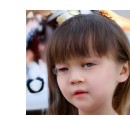 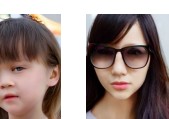 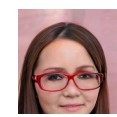 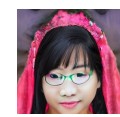 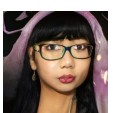

*Figure 19.* StyleGAN3 Outsourced Diffusion

## D.4. Bald man with black beard.

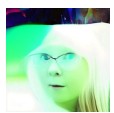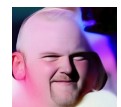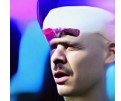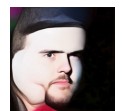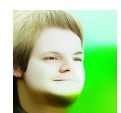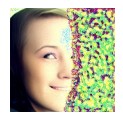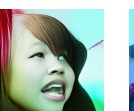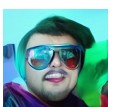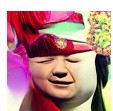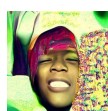

*Figure 20.* NVAE HMC

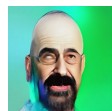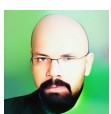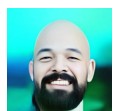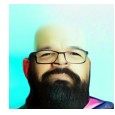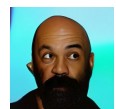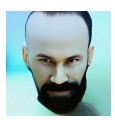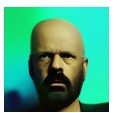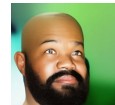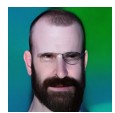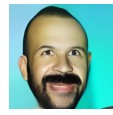

*Figure 21.* NVAE Outsourced Diffusion

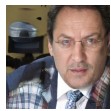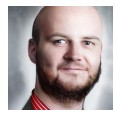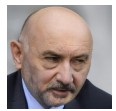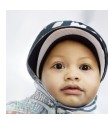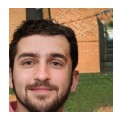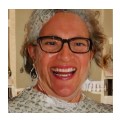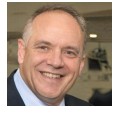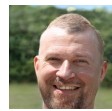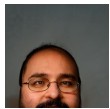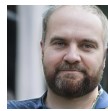

*Figure 22.* StyleGAN3 HMC

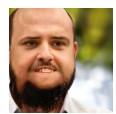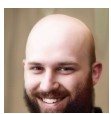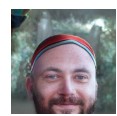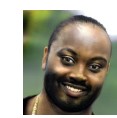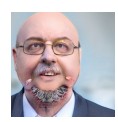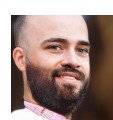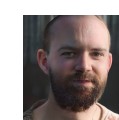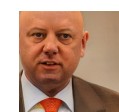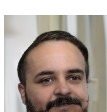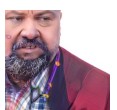

*Figure 23.* StyleGAN3 Outsourced Diffusion

## D.5. Brown haired child.

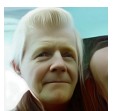 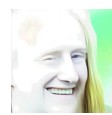 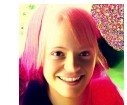 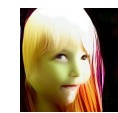 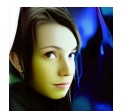 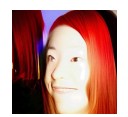 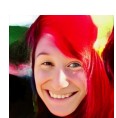 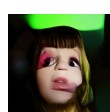 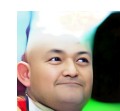 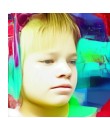

*Figure 24.* NVAE HMC

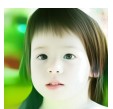 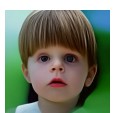 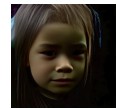 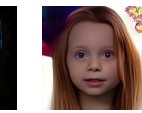 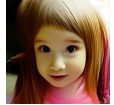 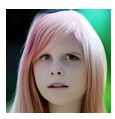 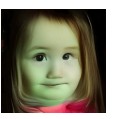 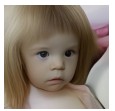 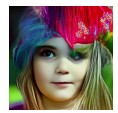 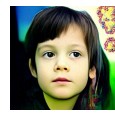

*Figure 25.* NVAE Outsourced Diffusion

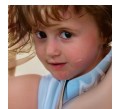 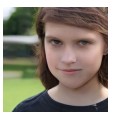 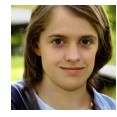 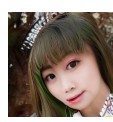 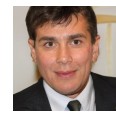 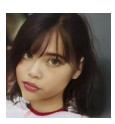 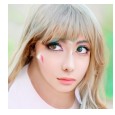 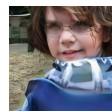 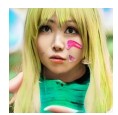 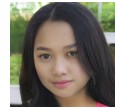

*Figure 26.* StyleGAN3 HMC

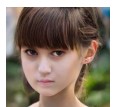 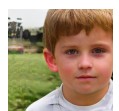 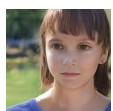 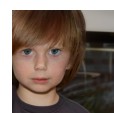 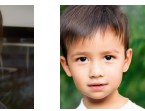 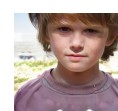 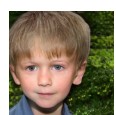 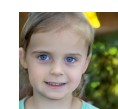 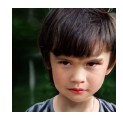 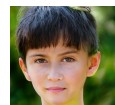

*Figure 27.* StyleGAN3 Outsourced Diffusion

# E. Stable Diffusion 3 Samples

## E.1. A cat and a dog.

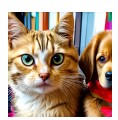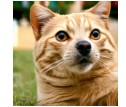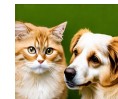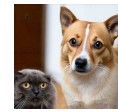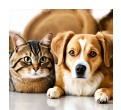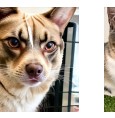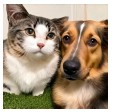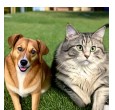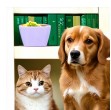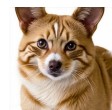

*Figure 28.* Prior

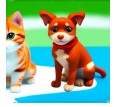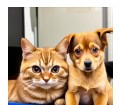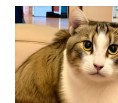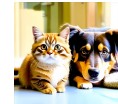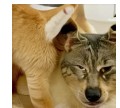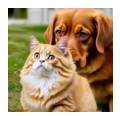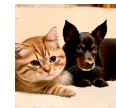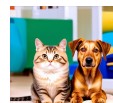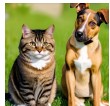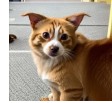

*Figure 29.* Classifer-Free Guidance

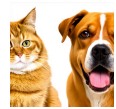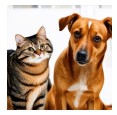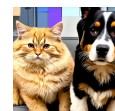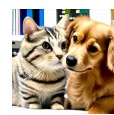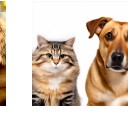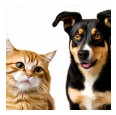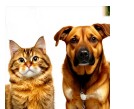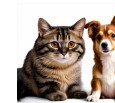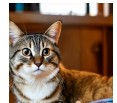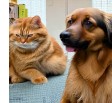

*Figure 30.* Outsourced Diffusion

## E.2. A cat riding a llama.

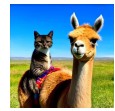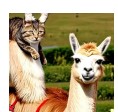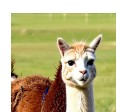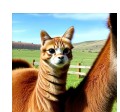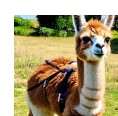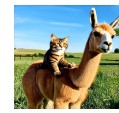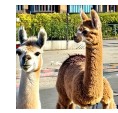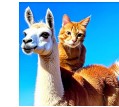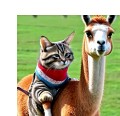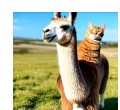

*Figure 31.* Prior

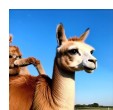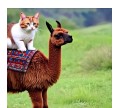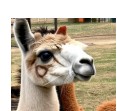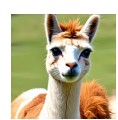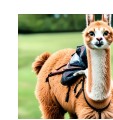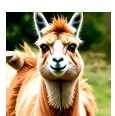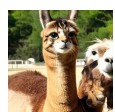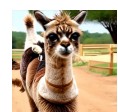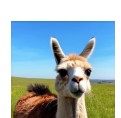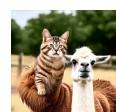

*Figure 32.* Classifer-Free Guidance

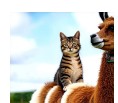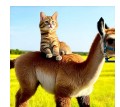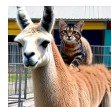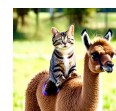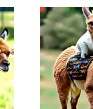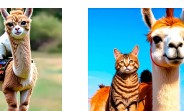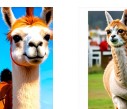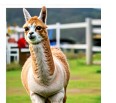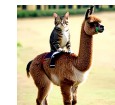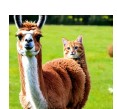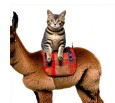

*Figure 33.* Outsourced Diffusion

### E.3. A quiet village is disrupted by a meteor strike.

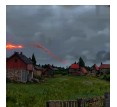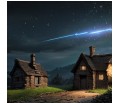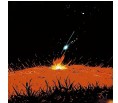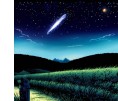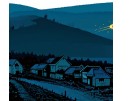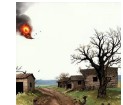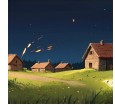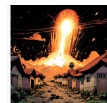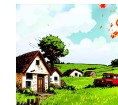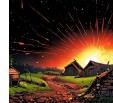

*Figure 34.* Prior

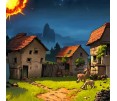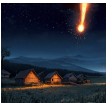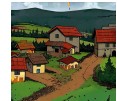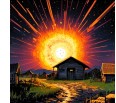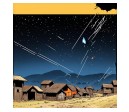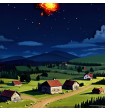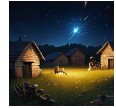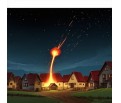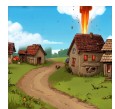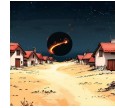

*Figure 35.* Classifer-Free Guidance

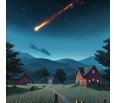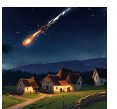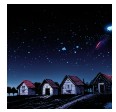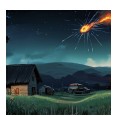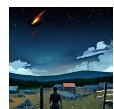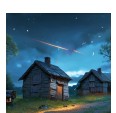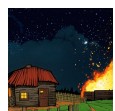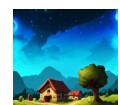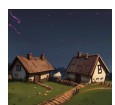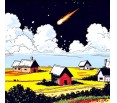

*Figure 36.* Outsourced Diffusion

### E.4. A human with a horse face and a human with a wolf face.

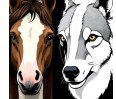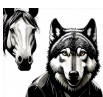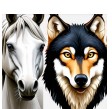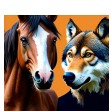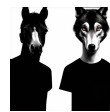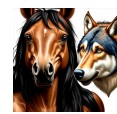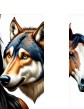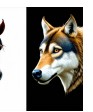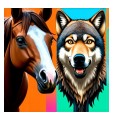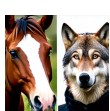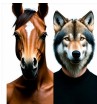

*Figure 37.* Prior

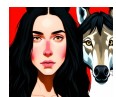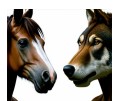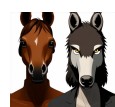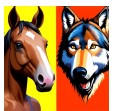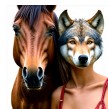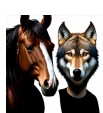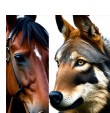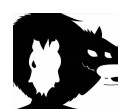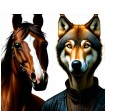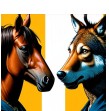

*Figure 38.* Classifer-Free Guidance

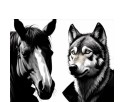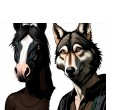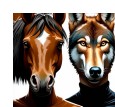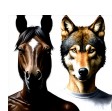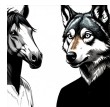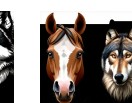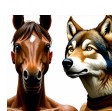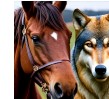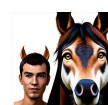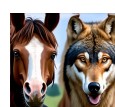

*Figure 39.* Outsourced Diffusion

## F. CIFAR-10 HMC Chain

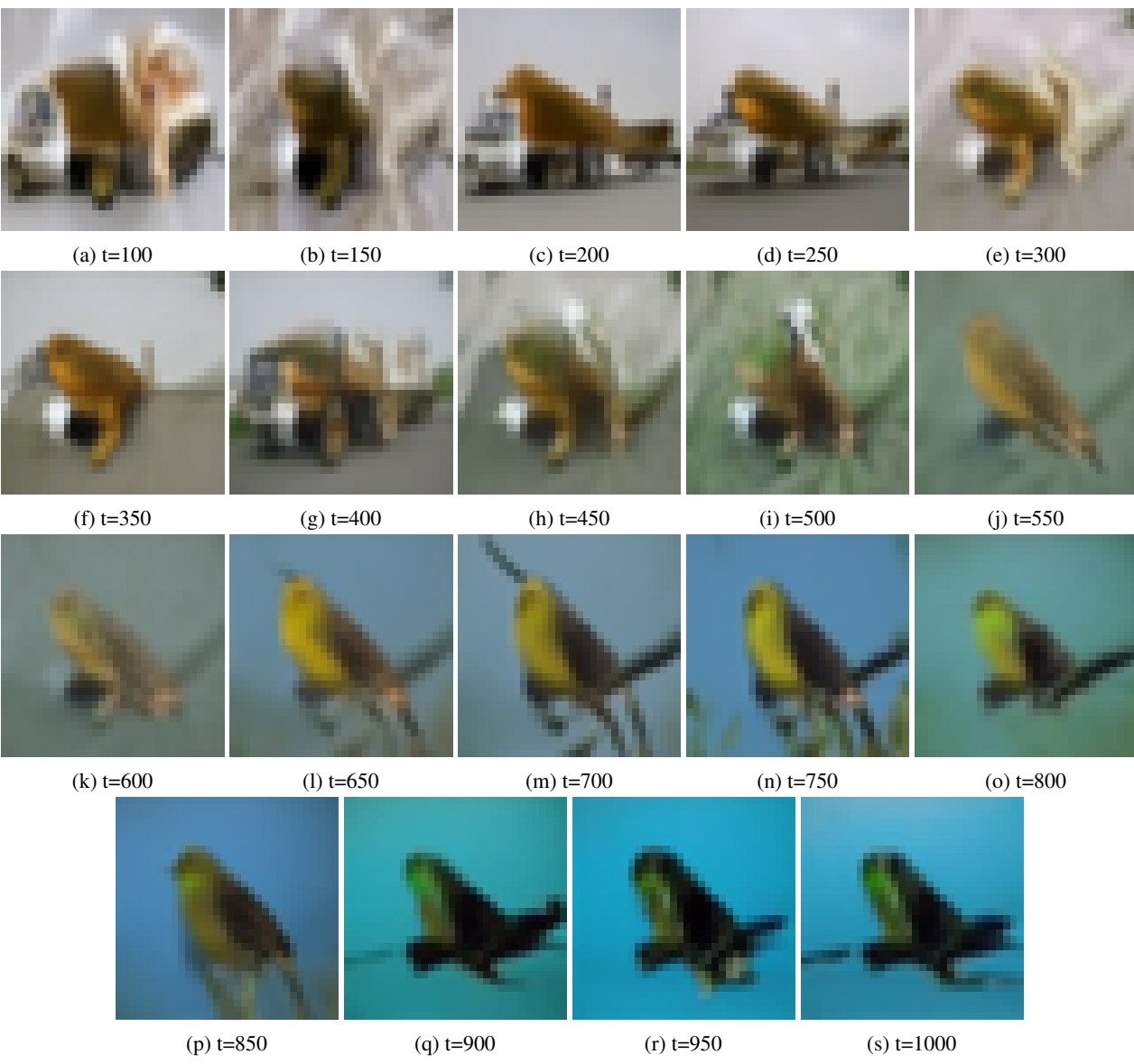

(a) t=100    (b) t=150    (c) t=200    (d) t=250    (e) t=300

(f) t=350    (g) t=400    (h) t=450    (i) t=500    (j) t=550

(k) t=600    (l) t=650    (m) t=700    (n) t=750    (o) t=800

(p) t=850    (q) t=900    (r) t=950    (s) t=1000

*Figure 40.* Samples from HMC chain of length 1000 after 100 steps of burn-in, for CIFAR class 'Airplane' with CNF prior. We find that in the latent space, MCMC smoothly traverses through different modes. The samples **(d,o,q,r,s)** are distinctly identifiable as airplanes and **(c)** is partially identifiable as an airplane. The samples approach the correct class after long mixing time.

