# OpenReview forum: "Outsourced Diffusion Sampling: Efficient Posterior Inference in Latent Spaces of Generative Models"
_ICML.cc/2025/Conference — ICML 2025 poster_

### Official Review · Reviewer_RBkw · 2025-03-08

**Overall Recommendation:** 4

**Summary:**

This paper presents a new method for posterior sampling from a wide variety of generative models (GANs, Flow, VAE) that can be expressed as a deterministic transformation x ~ f(z) for z sampled from a simpler distribution like Gaussian noise. The key idea is to train an auxiliary diffusion model to produce an initial point z' which when passed through f(z') results in samples x' from the the desired posterior in the data space p(x | y).

**Claims And Evidence:**

There are 3 main claims: 1) It's applicable to a wide range of prior models 2) It's effective under a variety of domains 3) It's more efficient than MCMC type methods.

I think the claims are well supported. Results are shown on a variety of models and a lot of different tasks. One thing that wasn't clear to me is whether we need to train a different model for each posterior task. If so, then the efficiency is questionable, because MCMC posterior sampling methods can generally substitute in different likelihood constraints p(y | x) at inference without retraining. Since the R constraint is in the loss function (eq 4) it seems like the outsourced diffusion model is trained for each specific task.

**Essential References Not Discussed:**

N/A

**Experimental Designs Or Analyses:**

The broad experiments were great. There were many different generative models, constraints, and datasets used. As mentioned earlier, it would be nice to see some higher resolution examples if it's possible.

**Methods And Evaluation Criteria:**

In general the method is well explained and makes sense. I like some of the core evaluations on the ImageReward examples to show how this proposed method improves over the baselines. I wonder why the examples are mostly 256 x 256. Is it harder to train in the latent space of larger models? Given that the first couple of figures are on toy examples (the swiss roll pictures) and the experiments are on small images, it seems like there may be more work needed to make this method truly applicable.

**Other Comments Or Suggestions:**

I think there should be one nice teaser figure at the top. To start the paper by only seeing the toy swiss roll examples makes me think the practicality is already limited. The casual reader would greatly benefit from seeing a clear, compelling example up front, like maybe the cat on the llama example.

**Other Strengths And Weaknesses:**

The paper is very clear. I found the method section easy to follow. I worry about the cost of training an auxiliary model for posterior sampling. A major advantage of using generative models for posterior sampling, is the ability to handle many tasks (inpainting, super-resolution etc) with a single pretrained generative model. If the outsourced diffusion sampler has to be retrained for different R functions then this method becomes much more constrained. In fact I think this factor should be discussed more, the (retraining of the outsourced sampler for different constraints)

**Questions For Authors:**

N/A

**Relation To Broader Scientific Literature:**

I think this paper ties together posterior sampling across many different types of generative models nicely. It also positions itself in contrast to MCMC posterior sampling methods, which have generally been extremely slow and required many steps to converge. I like that it is positioned well regarding all the posterior methods that the community uses.  It is also a novel idea from my understanding.

**Theoretical Claims:**

I checked most of the math closely (excluding some of the proofs in the appendix). Everything seemed to follow logically, and I appreciate the authors interpreting many of the detailed equations in more intuitive/layman terms.

---

> ### Author Rebuttal · Authors · 2025-04-01
>
> We thank the reviewer for their detailed review and constructive feedback, as well as the positive comments on the exposition and evaluations.
>
> To address the questions:
>
> ### __Efficiency of training a model for each task__
>
> It is true that the outsourced diffusion sampler must be trained separately for each likelihood constraint $p(y\mid x)$. However, for the diffusion sampler, the inference-time cost per sample is much lower than for MCMC (i.e., the sampling cost is amortized). In fact, the total cost for generating samples for evaluation was much lower using our method (**including** training time) compared to MCMC, for instance:
> - The CIFAR-10 evaluation required generating 1000 samples, which took 10 hours with Hamiltonian Monte Carlo (using multiple chains in parallel), while the outsourced diffusion sampler took 5 hours for both training and sampling (of which sampling was a neglible fraction).
> - A similar pattern held for MCMC in the protein experiments: the entire experiment took 8 hours for MCMC and 4 total hours for our method.
>
> Additional details on the timing comparisons between MCMC baselines and our method are available in Appendix B.
>
> Regarding the **need to retrain for new constraints**, we point out that the regular functionality of generative models remains intact when using our method to sample the latent space. Namely, we can still apply *approximate* training-free methods for guidance to solve intractable sampling problems (e.g., for inpainting or linear inverse problems), with the only difference being that the noise components are obtained from our diffusion sampler. Our method offers a computational trade-off whereby additional compute can be spent to sample more accurately according to a given posterior distribution, compared to the generative model with standard noise.
>
> ### __Size of example images: Can we use larger models?__
>
> The largest latent space we explore in our experiments is with Stable Diffusion 3, featuring a latent resolution of $16 \times 64 \times 64$, corresponding to an image resolution of $3 \times 512 \times 512$. We consider this to be a reasonably high resolution for practical applications: most image generative models, save those with excessive hardware and sampling time requirements, use similar latent dimensions. (Although diffusion models trained in pixel space with higher dimension exist, they typically require far more sampling steps than ones that work in a latent space.) Generally, larger latent spaces do increase training times, but our experiments demonstrate that our method scales effectively even with high-fidelity models like SD3.
>
> We note that for many generative models like GANs, the noise space is of much lower dimension than the data space (e.g., 512 vs. $3\times256\times256$ for the FFHQ StyleGAN3 we consider), which is further motivation to perform outsourced sampling.
>
> Finally, thank you for your suggestions on reorganizing the figures. We will incorporate this feedback into our final submission.
>
> **Thank you again for your review, and please do not hesitate to let us know if there is anything more we can clarify in the second response phase.**

---

> > ### Comment · Reviewer_RBkw · 2025-04-04
> >
> > thank you for your detailed responses. I didn't realize the images were 512 x 512. This is very reasonable resolution, and should be made clear by including a larger figure earlier on. I maintain my recommendation of 4 (accept).

---

### Official Review · Reviewer_Ge1S · 2025-03-09

**Overall Recommendation:** 3

**Summary:**

This paper addresses the posterior inference problem using diffusion sampling. By comparing their approach with existing MCMC methods and amortized inference methods, the authors demonstrate that their proposed outsourced diffusion sampling method, optimized through the trajectory balance objective, is both efficient and effective. They evaluate its performance across three application domains: conditional image generation, text-to-image generation, and protein structure generation.

**Claims And Evidence:**

This paper makes the following three claims. First, ODS is agnostic to the form of priors, which is demonstrated through the experiments. Second, ODS is an effective posterior inference method that can be applied across multiple domains. While ODS shows improved results in certain application domains given the baseline authors provided, some important baselines have not been considered, which I will discuss in detail in the Experiments & Designs section.

In addition, certain results are a bit far from the state-of-art methods, i.e. the FID of ODS with I-CFM in the conditional image generation task. The experiments cover three application domains, which I consider sufficient.

The third claim concerns efficiency, where the authors claim that ODS is more efficient than amortized inference methods and MCMC methods.  I suggest the authors make this more explicit, is it the training time efficiency or sampling time efficiency?
Training time ODS is better than Adjoint matching (Domingo-Enrich et al., 2024). But the NFEs at sampling time are not mentioned between ODS and Adjoint matching in the conditional CIFAR-10 data.

**Essential References Not Discussed:**

The discussion on the relevant works is sufficient.

**Experimental Designs Or Analyses:**

Class-Conditional Sampling with Cifar-10, Text-to-Image RLHF is carefully checked.

For the first one, a simple training-free baseline such as classifier-guidance is missing from Table 3. Additionally, I'm unsure if mentioning distillation on ODS is relevant.

For the second one, is the choice of SD3 instead of SD1.5 as the prior for not comparing with Fan et al. (2023) and Venkatraman et al. (2024)? Both methods are closely related to the proposed method. Although the paper mentions that CNF is not a diffusion model, recent work [1] demonstrates that flow-based models can be viewed as diffusion models with different noise schedules and parameterizations. Therefore, comparisons should also include methods closely related to this work in the diffusion land.


[1] Gao, R., Hoogeboom, E., Heek, J., De Bortoli, V., Murphy, K. P., & Salimans, T. (2024). Diffusion meets flow matching: Two sides of the same coin.

**Methods And Evaluation Criteria:**

The proposed methods is a proper fit to the problem and also to the applications that the author is trying to target.
In the Conditional High-Resolution Face Generation and Text-to-Image RLHF experiments, the generated image quality metric like FID is not present.

**Other Comments Or Suggestions:**

Figure 1 comment: “The constraint function – a mixture of two Gaussians centered **an an** observation”

**Other Strengths And Weaknesses:**

- Strength:

The proposed method offers a promising solution to the posterior inference problem at scale and is universal to the prior distribution. In addition, the proposed method does not require the reward or constraint function to be differentiable. The paper also includes applications from different domains.

- Weakness

My main concerns in this work are the writing and the experiment. The details of the proposed method in Section 4.2 are missing, and there is no algorithm block describing the overall method. This leaves an audience with no prior knowledge about the trajectory balance objective line of work struggling to differentiate between what already exists in the literature and what is newly proposed in this work. So I suggest instead of introducing what VAE/GAN is, set up the audience with more details like “off-policy” divergences approach and TB objective.

Some of the weaknesses are mentioned in the Experimental Designs or Analyses. Some of the results, such as the FID scores on conditional CIFAR-10, are far from those of state-of-the-art methods, which makes hard to judge whether ODS can obtain high quality posterior samples.

**Questions For Authors:**

In section 4.2, which part of the proposed method exists in the literature and which part is new in this work?

Did authors consider training-free posterior sampling methods to be a proper baseline?

Why is classifier guidance not appear in the condition Cifar10 experiment?

Could the author compare ODS with Fan et al. (2023) and Venkatraman et al. (2024) on SD1.5 to demonstrate the effectiveness?

Is the training stable given the ODS need to back-propogate through the entire sampling chain according to Eq(4)?

**Relation To Broader Scientific Literature:**

Posterior inference is an important problem to the broader scientific field. Traditional MCMC methods may yield better results, but they suffer from slow sampling times. Amortized inference offers an alternative approach, however, it requires obtaining massive numbers of samples from the posterior for training. Therefore, developing an efficient solution to this problem is important, which is what this paper targets.

**Theoretical Claims:**

N/A

---

> ### Author Rebuttal · Authors · 2025-04-01
>
> We appreciate your good questions and constructive feedback. We have responded to your points below.
>
> ### __Algorithm details__
>
> We agree that the addition of an algorithm block would help readers understand how to implement the core training loop, and added this to __Algorithm 1 in page 2 of the [linked pdf](https://anonymous.4open.science/r/results-FED2/results.pdf).__
>
> ### __FID scores not state-of-the-art__
>
> While FID scores are indeed worse than adjoint matching on CIFAR-10, we highlight that ODS is a general approach applicable to any generative model, while adjoint matching is specific to flow models (and not all flow models, see end of §3.1) and requires differentiating the reward.
>
> The FID scores reported in [Venkatraman et al.] are an order of magnitude lower than those presented in our paper. On examining the underlying codebases we discovered that the discrepancy arises from the FID packages used. Specifically, [Venkatraman et al.] uses [this codebase](https://github.com/marcojira/fld), while we use [this one](https://github.com/GaParmar/clean-fid). The distinction lies in the handling of image resolution: CIFAR images are smaller than ImageNet images, and only the latter implementation (which we use) accounts for this difference appropriately.
>
> ### __Efficiency__
> Our efficiency claim refers to lower training time of ODS relative to finetuning methods, while having much cheaper inference than MCMC. There is a small (25 step) inference overhead relative to Adj. matching (for reducing this, see Appendix C.1).
>
> ### __Which part of the proposed method is new?__
>
> Trajectory balance has previously been employed for training diffusion samplers from unnormalized densities (notably in [Lahlou et al.] and [Sendera et al.]), and a variant of it, relative TB, is a diffusion-specific fine-tuning method in [Venkatraman et al.]. Our novel contribution is the general capability to train a sampler (using TB) of the Bayesian posterior in noise space (§3.2) for any generative model, providing a general-purpose framework for amortized posterior inference. The exact way in which ODS generalizes relative TB is detailed in Appendix A.1.
>
> ### __Training-free baselines and classifier guidance__
>
> Unbiased **classifier guidance** requires a *time-dependent* likelihood gradient: $\nabla \log p(y \mid x_t) = \nabla \log \mathbb{E}_{p(x_0 \mid x_t)}[p(y \mid x_0)]$
>
> This necessitates training a classifier on noised inputs, as using one trained on clean data renders the gradient intractable. Such methods therefore make stronger assumptions than our setting, which assumes a black-box classifier on clean data.
>
> Training-free methods like DPS are biased approximations to classifier guidance but work well in practice. DPS uses the approximation $\nabla \log p(y \mid x_t) \approx \nabla \log p(y \mid \mathbb{E}[x_0 \mid x_t])$, and requires the same assumptions as our proposed algorithm making it an appropriate baseline.
>
> We’ve added results for DPS (Diffusion Posterior Sampling, [Chung et al.]), a strong training-free baseline for approximate posterior sampling with diffusion and flow models using the I-CFM prior on CIFAR-10. __Results are in Table 1 of the response to Reviewer qmBQ and also [here](https://anonymous.4open.science/r/results-FED2/results.pdf).__ DPS achieves high reward but poor FID, reflecting highly biased sampling.
>
> ### __Additional experiments with SD1.5__
>
> We conducted additional experiments with Stable Diffusion 1.5 to compare against DDPO [Black et al.], DPOK [Fan et al.], and RTB [Venkatraman et al.], as requested. We also included RTB with the I-CFM prior as a baseline for the CIFAR-10 class-conditional sampling task. __Results are shown in Tables 2 and 1 of our response to reviewer qmBQ and are available [here](https://anonymous.4open.science/r/results-FED2/results.pdf).__ On SD1.5, ODS achieves the a strong balance between reward and sample diversity.
>
> ### __Stability of training__
>
> ODS uses TB, an off-policy RL objective that does **not** require backpropagating through the full sampling chain. Instead, it only needs local gradients of log-likelihoods for each sampling step (see [Venkatraman et al.], Appendix H.1)
>
> Such off-policy optimization has three benefits, according to prior work (see, e.g., [Nüsken & Richter], [Sendera et al.]):
> - The reward function need not be differentiable, since TB avoids using log-reward gradients;
> - TB avoids instability and mode collapse seen in methods that backprop through the full SDE trajectory (e.g., PIS [Zhang & Chen]);
> - Memory usage is minimal thanks to a gradient accumulation strategy (included in our code), avoiding storage of full computation graph
>
> In our experiments, we found training to be very stable for all tasks except for the protein diversity task, where policy collapse sometimes occurs.
>
> **Thank you again for your review, and please do not hesitate to let us know if there is anything more we can clarify in the second response phase.**

---

### Official Review · Reviewer_nz5P · 2025-03-09

**Overall Recommendation:** 4

**Summary:**

This work targets the problem of generating samples from the posterior $p(x | y) \propto p(x) r(x, y)$, where the prior $p(x)$ is a (pre-trained) generative model and $r(x, y)$ is a reward function. The authors argue that (most) generative models can be formulated as the application of a pushforward $f(z) = x$ to a simple latent distribution $p(z)$ (e.g. Gaussian). In this case, the initial problem is equivalent to generating from the posterior $p(z | y) \propto p(z) r(f(z), y)$, which the authors argue is easier to tackle. The authors train a *diffusion sampler* $p^\phi(z | y)$ using the trajectory balance (TB) objective (Malkin et al., 2022), which only requires access to the unnormalized density $p(z) r(f(z), y)$. The authors demonstrate on 3 text-to-image benchmarks and 1 protein design benchmark that their method is competitive against a few alternatives.

### update after rebuttal

I thank the authors for their rebuttal and taking my comments into account. I appreciate the additional discussion, and hope that they will be able to add it to the manuscript. My concerns regarding the evaluation of the posterior (calibration) somewhat remains, but I agree that proper evaluation remains a challenge in high dimension, especially in the lack of data pairs $(x, y)$. In this light, I will be raising my score to 4 (accept).

**Claims And Evidence:**

/

**Essential References Not Discussed:**

There is an extensive literature on the evaluation of posterior distributions [3-5] (this is a very limited selection), which this work does not consider/discuss.

**Experimental Designs Or Analyses:**

See Methods and Evaluation criteria.

**Methods And Evaluation Criteria:**

Yes, the method is sound, the evaluation tasks are relevant, and several baselines are considered. However,

1. In the case where $p(x)$ is a flow-matching/diffusion model, an important baseline should be to fine-tune the generative model using the trajectory balance (TB) objective directly, which would validate the authors' claim that $p(z | y)$ is an easier target than $p(x | y)$ (line 74.5). This approach is taken by Venkatraman et al. [1].

2. The assessment of the quality of the inferred posterior distributions is not sound. $\mathbb{E}[\log r(f(z), y)]$ is maximized by a collapsed distribution around $z^* = \arg\max_z r(f(z), y)$. The diversity is maximized by a maximal entropy distribution. Although these quantities are relevant for evaluating the inferred posterior $p^\phi(z | y)$, they are not sufficient. In fact, I suspect the method presented in this paper to be subject to mode collapses, and the current evaluation does not rule out this hypothesis. For example, you can observe in Figure 4.e that most dogs are white and facing the camera, which is not the case in the CIFAR-10 dataset. Similarly, in Figure 6, there is a collapse of the image composition (tabby cat on the llama, meteor falling bottom left, horse on the left).

    There is an extensive literature on the evaluation of posterior distributions, notably emerging from the SBI [2] community. For example, in scientific applications, the calibration of the posterior distribution is extremely important [3-5]. Another sensible metric could be the (reverse) KL divergence between $p^\phi(z | y)$ and $p(z | y)$. This KL can be computed up-to the normalizing constant $Z(y) = \int p(z) r(f(z), y) dz$, and therefore can be used to compare different approximations of $p(z | y)$.

    In my opinion, a proper evaluation of the inferred posteriors is a *sine qua non* for a paper claiming to perform posterior inference. I would be happy to raise my score if the authors address this concern.

[1] Amortizing intractable inference in diffusion models for vision, language, and control (Venkatraman et al., 2024)

[2] The frontier of simulation-based inference (Cranmer et al., 2020)

[3] Validating Bayesian Inference Algorithms with Simulation-Based Calibration (Talts et al., 2018)

[4] A Trust Crisis In Simulation-Based Inference? Your Posterior Approximations Can Be Unfaithful (Delaunoy et al., 2021)

[5] Sampling-Based Accuracy Testing of Posterior Estimators for General Inference (L3mos et al., 2023)

**Other Comments Or Suggestions:**

1. In Table 1, diffusion models should have $d_{data}$ as noise dimension, if they are to be considered as deterministic functions. The decoder of latent diffusion models is typically considered deterministic (even by the authors), leading to a noise dimension $d_{latent}$. This is consistent with Table 2.
1. In Figure 1, "two Gaussians **and** an observation"
1. In Figure 1, the bottom right plot should read $p(x | y)$
1. Line 240, "**the one** defined by the target"
1. Line 241, "noising kernel" is undefined
1. Line 246, TB "loss" was introduced as "objective" in Eq. (4)
1. In Eq. (2), I would avoid the notation $\sigma_t$ as it has another meaning in the diffusion model literature
1. Line 312.5 $z \in \mathbb{R}^{128}$

**Other Strengths And Weaknesses:**

1. The paper is very well written. The figures are clear. The experiments are relevant.
2. The evaluation is lacking a proper assessment of the posteriors' quality.
3. In my opinion, the methodological contribution of this work is modest with respect to previous works. The impact, however, could be significant. Unfortunately, this cannot be assessed without a proper evaluation.

**Questions For Authors:**

none

**Relation To Broader Scientific Literature:**

The literature review in this work is complete and extensive, without being overwhelming. Kudos to the authors.

**Theoretical Claims:**

Yes, I quickly checked the proof of Prop. 3.1 and it seemed valid. The work is mainly applicative, as it consists in applying already existing methods (mainly TB) to a widespread problem.

---

> ### Author Rebuttal · Authors · 2025-04-01
>
> Thank you for the detailed review and constructive feedback, as well as pointing out that the paper is very well written.
>
> You raise some important concerns, which have helped us improve the paper and which we hope to address:
>
> ### __Posterior evaluation__
>
> We agree that more comprehensive posterior evaluations are important for a thorough assessment.
>
> First, we emphasize that the FID scores reported in our original CIFAR-10 experiments capture both reward and coverage/diversity, and serves as a proxy metric for closeness to the true posterior. FID remains a widely accepted and robust metric for evaluating the sample quality of image generative models.
>
> For the CIFAR-10 experiments (comparing our amortized sampler to ground truth posterior samples), we tried posterior evaluation with TARP [Lemos et al., 2023] and PQMass [Lemos et al., 2024], an unconditional two-sample variant of TARP, but were not able to obtain results that seemed meaningful. For example, running TARP frequently showed prior samples as perfectly calibrated and unbiased relative the conditional dataset, which is obviously false -- but the discrepancy is captured correctly in the FID scores. (Published code from those papers was used; to ensure the results do not contain an error, we checked that our code could reproduce values in line with those reported on CIFAR-10 in [Lemos et al., 2024].)
>
> For other experiments, ground truth samples from the posterior are not available, complicating evaluation (for instance, FID cannot be computed), which is the reason we resort to other metrics. For prompt-conditioned sampling we instead report the average CLIP feature distance, following the approach used by [Venkatraman et al.]. In our Stable Diffusion 1.5 experiments, the CLIP feature diversity scores exhibit a consistent and interpretable trend: the prior achieves the highest diversity, while DDPO, which performs greedy reward maximization without KL regularization, yields the lowest diversity. These trends support the reliability of our evaluation metrics in capturing meaningful aspects of sample diversity.
>
> At the reviewer's suggestion, we also report the ELBO (i.e., the lower bound on the log-normalizing constant $\log Z$, where the ELBO-to-log-likelihood gap is the reverse KL divergence). This is a widely used metric in the diffusion samplers literature, including [Sendera et al.], and is maximized by a perfect sampler of the target density. We note however, that this metric cannot be used to evaluate inference-time baselines or MCMC.
>
> __Addition of the ELBO metrics for CIFAR-10 and SD3 are included in Tables 1 and 3 repectively, within the [linked PDF](https://anonymous.4open.science/r/results-FED2/)__.
>
> ### __Additional baselines__
>
> We include experimental results for the RTB baseline proposed by [Venkatraman et al.] for flow models trained with independent coupling, which can subsequently be fine-tuned as diffusion models. This evaluation is done for RTB using the I-CFM prior on CIFAR-10; we also include new results comparing our method against RTB, DPOK [Fan et al.] and DDPO [Black et al.] with Stable Diffusion 1.5 as a prior with the same setup used by [Venkatraman et al.].
> __These are included in Tables 1 and 2 of the response to Reviewer qmBQ, and also in the [linked pdf](https://anonymous.4open.science/r/results-FED2/results.pdf)__.
>
> We note that for CIFAR-10, the RTB baseline is significantly more unstable to train: requiring LoRA fine-tuning [Hu et al.], as well as lower learning rates to avoid a quick policy collapse. This aligns with design choices in the RTB paper [Venkatraman et al.].
> Even with this, the training remains somewhat unstable for the flow model architecture, being very sensitive to training hyperparameters. Within the training time allocated to ODS, the model achieves a modest improvement in reward; however, the generated samples often fail to consistently belong to the target class and frequently exhibit visual artifacts. This degradation in quality is further reflected in poorer posterior metrics (FID, ELBO). We argue the training instability is caused by the flow prior, which uses a different noise schedule as well as much fewer inference steps compared to experiments in [Venkatraman et al.]. This instability could lead to "reward-hacking" behavior where reward improves at the cost of sample quality.
>
> By contrast, ODS improves largely within the first 2-3 GPU hours, and only marginally afterwards. The general difficulty of training RTB with the flow prior reinforces the claim that the latent posterior $p(z|y)$ is a simpler target than the image posterior $p(x|y)$.
>
> Finally, **thank you for your helpful suggestions on improving the notation and writing clarity**, as well as for pointing out the typos. We will implement this feedback for our final submission.
>
> **Thank you again for your review, and please do not hesitate to let us know if there is anything more we can clarify in the second response phase.**

---

### Official Review · Reviewer_qmBQ · 2025-03-13

**Overall Recommendation:** 3

**Summary:**

## Summary
* This paper proposed a more general approach towards posterior inference for generative model with Gaussian prior (e.g. GAN, Flow, Diffusion models). More specifically, it proposed to learn a non-Gaussian prior (noise space z).
* The paper is evaluated on various models and various scale from toy size to Stable Diffusion 3. Solid empirical results support

**Claims And Evidence:**

Yes. The authors claim a general approach to achieve conditional sample by learning the prior p(z) of various generative model. They verify their approach in different kinds of models such as flow and diffusion.

**Essential References Not Discussed:**

See Relation To Broader Scientific Literature.

**Experimental Designs Or Analyses:**

## Experimental designs
* The flow matching model in SD3 is a special type of flow matching model that start from a Gaussian noise. This type of flow is not much different from diffusion models
* For some models such as GAN and diffusion, there exists tailored method to sample from posterior, such as DPS [diffusion posterior sampling for general noisy inverse problems] and GAN inversion [PULSE: Self-Supervised Photo Upsampling via Latent Space Exploration of Generative Models]. As the authors propose a general approach for different types of generative model, it would strengthen the paper if the authors show the performance of their approach against model specific approaches.

**Methods And Evaluation Criteria:**

The evaluation spans from toy CIFAR dataset to practical scale image generation. The metrics can be organized in a better way, but common metrics such as FID and imagereward are shown.

**Other Comments Or Suggestions:**

-

**Other Strengths And Weaknesses:**

## Weakness
* The authors are encouraged to organize the results in a better way. For now too many results are shown from image to portein generation. Different datasets, methods and metrics are scattered across the main text and appendix.

**Questions For Authors:**

-

**Relation To Broader Scientific Literature:**

* The authors propose a general approach to learn amortized conditional generative model by learning the prior. This paper is also related to a recent hot topic: golden noise [Not All Noises Are Created Equally:Diffusion Noise Selection and Optimization] and test time scaling in text to image diffusion model [Inference-Time Scaling for Diffusion Models beyond Scaling Denoising Steps]. Despite this paper is amortized, it is better to discuss the relationships to those non-amortized works.

**Theoretical Claims:**

There is no theoretical claim in this paper.

---

> ### Author Rebuttal · Authors · 2025-04-01
>
> Thank you for the helpful review. We address the weaknesses and questions raised by you in our response below.
>
> ### __Comparison with model-specific approaches__
>
> We thank the reviewer for pointing out the additional model-specific baselines with which to compare out general approach.
> - We note that for GANs, we compare to a latent space exploration baseline similar to PULSE, Hamiltonian Monte Carlo (HMC).
> - We have added the DPS baseline to the CIFAR-10 experiments.
> - We have added RTB as a diffusion-specific baseline for CIFAR-10 and Stable Diffusion-1.5 experiments.
>
> __The additional results are included in Tables 1 and 2 below, and also in the [linked pdf](https://anonymous.4open.science/r/results-FED2/results.pdf).__
>
> Table 1: CIFAR-10 posterior sampling results, averaged over 10 classes
> | Model | Sampler              | $\mathbb{E}[\log p(\textbf{y} \mid \textbf{x})]$ $(\uparrow)$ | FID $(\downarrow)$ | ELBO $(\uparrow)$ |
> |-------|----------------------|------------------------------------------------------------|--------------------|--------------------------------|
> | I-CFM | Prior                | -5.88                                                      | 84.79              |      -24.04                    |
> |       | DPS                  | -2.22                                                      | 84.96              |        -                       |
> |       | RTB                  | -4.20                                                      | 90.77              |    -147.69                     |
> |       | Latent HMC           | -2.80                                                      | 46.69              |        -                       |
> |       | Adj. Matching        | -3.09                                                      | 19.45              |      -17.23                    |
> |       | **Outsourced Diff.** | -3.35                                                      | 34.28              |      -20.36                    |
>
> In the CIFAR-10 results we note that DPS obtains a higher reward than Outsourced Diff., with a worse FID score (and ELBO -- see the response to Reviewer nZ5P). This points to possible reward hacking. Additionally, the added RTB baseline for this experiment proved much more unstable during training than our method, reflected in its worse FID (and ELBO) scores, while improving the reward relative to the prior. Some RTB runs underwent policy collapse, requiring early stopping for the reported results.
>
> Table 2: SD 1.5 fine-tuning results. DDPO, DPOK and RTB results taken from [Venkatraman et al.]
>
> | Sampler              | $\mathbb{E}[\log r(\textbf{x}, \textbf{y})] (\uparrow)$ | CLIP diversity $(\uparrow)$ |
> |----------------------|--------------------------------------------------------|-----------------------------|
> | Prior                |                -0.17                                   |          0.18               |
> | DDPO                 |                 1.37                                   |          0.09               |
> | DPOK                 |                 1.23                                   |          0.13               |
> | RTB                  |                 1.4                                    |          0.11               |
> | **Outsourced Diff.** |                 1.26                                   |          0.14               |
>
>
> For the SD1.5 experiment, we see that our method achieves a slightly lower reward than RTB, while obtaining a higher diversity over all baselines (other than the prior).
>
> ### __Relationship to non-amortized methods (golden noise and test-time scaling)__
>
> We agree that incorporating a discussion comparing our proposed amortized sampler with these inference-time optimization methods would strengthen the paper. Notably, the latent space HMC baseline used in our CIFAR-10 and FFHQ experiments shares conceptual similarities with these inference-time noise optimization approaches. While the methods in prior work do not asymptotically sample from the true Bayesian posterior over the noise space, they effectively bias the sampling process toward high-reward regions, often producing qualitatively similar results. In contrast, our method frames prior-regularized reward fine-tuning as posterior inference, whereas these inference-time techniques are more aligned with direct reward maximization. We will integrate this discussion into the appendix of the final version.
>
> ### __Organization of results__
>
> We thank the reviewer for their feedback on the organization of results. We will merge some of the results that share similar metrics and baselines (such as CIFAR and FFHQ), and reorganize the tables to make the presentation more clear for the final submission.
>
> **Thank you again for your review, and please do not hesitate to let us know if there is anything more we can clarify in the second response phase.**

---

### Decision · Program_Chairs · 2025-05-01

**Decision:**

Accept (poster)

**Comment:**

This paper proposes a method for sampling from data posteriors within the framework of large pretrained generative models, via diffusion samplers. The method is model-agnostic and can be applied to a variety of generative models that are deterministic push forward functions on Gaussian distribution. Experimental results have been shown on variety of tasks and models demonstrating the method's wide utility.

3/4 reviewers have recommended acceptance post rebuttal and the major concern of the reject review is on organization and writing.

I think this is a good paper worth accepting, albeit I urge the Authors to address the comments on better writing and evaluating the method against some state of the art models if possible.